

# Regularization methods for the combination of heterogeneous observations using spherical radial basis functions

Qing Liu[1], Michael Schmidt[1], Roland Pail[2], Martin Willberg[2]

[1]Deutsches Geodätisches Forschungsinstitut, Technische Universität München (DGFI-TUM), Arcisstr. 21, 80333 München, Germany
[2]Institute for Astronomical and Physical Geodesy, Technische Universität München, Arcisstr. 21, 80333 München, Germany

*Correspondence to*: Qing Liu (qingqing.liu@tum.de)

**Abstract.** Various types of heterogeneous observations can be combined within a parameter estimation process using spherical radial basis functions (SRBF) for regional gravity field refinement. However, this process is in most cases ill-posed, and thus, regularization is indispensable. We discuss two frequently used methods for choosing the regularization parameter which are the L-curve method and variance component estimation (VCE). Based on these two methods, we propose two new approaches for the regularization parameter determination, which combine the L-curve method and VCE.

The first approach, denoted as 'VCE + L-curve method', starts with the calculation of the relative weights between the observation techniques by means of VCE. Based on these weights the L-curve method is applied to determine the regularization parameter. In the second approach, called 'L-curve method + VCE', the L-curve method determines first the regularization parameter and it is set to be fixed during the calculation of the relative weights between the observation techniques from VCE. These methods are investigated based on two different estimation concepts for combining various observation techniques. All the methods are applied and compared in six study cases using four types of observations in Europe. The results show that the 'VCE + L-curve method' delivers the best results in all the six cases, no matter using SRBFs with smoothing or non-smoothing features. The 'L-curve method + VCE' also gives rather good results, generally outperforming the cases just using the L-curve method or VCE. Therefore, we conclude that the newly proposed methods are decent and stable for regularization parameter determination when different data sets are combined and can be recommended regardless of the type of SRBFs used.

## 1 Introduction

Gravity field modeling is a major topic in geodesy, and it supports lots of applications including physical height system realization, orbit determination and solid earth geophysics. To model the gravity field, a proper approach needs to be set up to represent the input data as good as possible. The global gravity field is usually described by spherical harmonics (SH), due to the fact that they are fulfilling the Laplacian differential equation and are orthogonal basis functions on a sphere. However, in the spatial domain, a global coverage of data sets is not always fulfilled sufficiently, and it is well-known that SHs cannot



represent data of heterogeneous density and quality in a proper way (Schmidt et al., 2007). Regional gravity refinement is, thus, performed for combining different observation types such as airborne, shipborne or terrestrial measurements which are only available in specific regions. Different regional gravity modeling methods have been developed during the last decades, e.g., the statistical method of Least Squares Collocation (LSC) (see Krarup, 1970; Moritz, 1980; Pail et al., 2010) or the method

of mascons (mass concentrations) (see Rowlands, 2005). The method based on spherical radial basis functions (SRBF) will be the focus of this work.

The fundamentals of SRBFs can be found amongst others in Holschneider et al. (2003) as well as Freeden and Michel (2004). Due to the fact that SRBFs are isotropic and characterized by their localizing feature, they can be used appropriately for regional approaches to consider the heterogeneity of data sources; examples are given by Marchenko et al. (2003), Schmidt et

al. (2007), Lieb et al. (2016). There are many factors in SRBF modeling that influence the accuracy of the regional gravity model, e.g., the shape, bandwidth, locations of the SRBFs and the extension of the data zone. Tenzer and Klees (2008) compared the performance of different types of SRBFs using terrestrial data, Bentel et al. (2013a, b) gave a comparison of nine different SRBFs in regional gravity field modeling based on simulated data. Bentel et al. (2013a) also studied the influence of point grids, and the results show that the differences between SRBFs are much more significant than the differences between

different point grids. Another detailed investigation about the locations of SRBFs can be found in Eicker (2008), where methods for choosing a proper bandwidth were also introduced. Bentel (2013a) discussed the reasons for edge effects and Lieb (2017) provided a possible way to choose area margins in order to minimize edge effects.

After setting up all the factors, heterogeneous data sets are often combined within a parameter estimation process; two combination models (see Schmidt et al., 2015) are introduced and applied in this work. One model takes the relative weightings

between the observation techniques into consideration while the other one relies on an equally weighted scenario. Regional gravity modeling is usually an ill-posed problem due to (1) the number of chosen basis functions, i.e. the SRBFs, (2) given data gaps, and (3) the downward continuation. Thus, regularization is in most cases inevitable in the parameter estimation process. Bouman (1998) discussed and compared different regularization methods, including Tikhonov regularization (Tikhonov and Arsenin, 1977), truncated singular value decomposition (TSVD, Xu, 1998), and iteration methods (Schuh,

1996). Choosing an appropriate regularization parameter is a crucial issue for a proper regularization. Generalized cross validation (GCV, Golub et al., 1979), L-curve criterion (Hansen, 1990; Hansen and Oleary, 1993) and variance component estimation (VCE, Koch, 1999; Koch and Kusche, 2002) are the three most commonly used methods for estimating the regularization parameter.

However, there are not many studies that compare the performance of each method, and the existing publications do not reach

an agreement indicating which gives the best, i.e., the most realistic results. Kusche and Klees (2002) compared GCV and the L-curve method, and the results show that the L-curve criterion always yields over-smoothed solutions; the same results were indicated by Xu (1998). Naeimi (2013) showed that the L-curve method provides satisfactory results while VCE and GCV cannot regularize the regional solutions sufficiently. Bentel (2013) presented that the L-curve method leads to fairly good results for noise-free data but does not perform as good as VCE in the case of noisy observations. Naeimi et al. (2015)



investigated how the performance of the regularization method changes when different types of SRBFs are used. The L-curve method delivers the best results when a non-smoothing kernel (Shannon) is applied, while the opposite when smoothing kernels are used. Besides, Lonkhuyzen et al. (2001) showed that the knowledge of the variances of the observations is not a guarantee for obtaining good solutions.

Thus, the purpose of this study is to find out the best-performing method for regularization parameter determination. We will (1) compare the performance of the L-curve method and VCE based on the aforementioned two combination models, (2) propose two new methods which combine the L-curve method and VCE together, (namely 'VCE + L-curve method' and 'L-curve method + VCE') and compare the results to the ones obtained using the L-curve method or VCE alone, and (3) test the stability of each method when different SRBFs are applied.

This work is organized as follows: in Section 2, we present the fundamental concepts of SRBFs, different types of gravitational functionals and SRBFs are also represented briefly. Section 3 discusses the parameter estimation, the Gauss-Markov model as well as the two combination models. Section 4 is dedicated to the regularization method, the L-curve method, VCE and the two newly proposed combination methods. In Section 5, the study area, the simulated data used in the study are presented as well as the results. The performance of all five methods for choosing the regularization parameter is compared. Finally, a

summary and conclusions will be given in Section 6.

## 2 Regional gravity field modelling using SRBF

In general, a spherical basis function $B(\boldsymbol{x}, \boldsymbol{x}_k)$ related to a point $P_k$ with position vector $\boldsymbol{x}_k$ on a sphere $\Omega_R$ with radius $R$ and an observation point $P$ with position vector $\boldsymbol{x}$ can be expressed by

$$B(\boldsymbol{x}, \boldsymbol{x}_k) = \sum_{n=0}^{\infty} \frac{2n+1}{4\pi} \left(\frac{R}{r}\right)^{n+1} B_n P_n(\boldsymbol{r}^{\mathrm{T}} \boldsymbol{r}_k) \, , \tag{1}$$

(Schmidt et al., 2007), with $\boldsymbol{x} = r \cdot \boldsymbol{r} = r \cdot [\cos\varphi \cos\lambda, \cos\varphi \sin\lambda, \sin\varphi]^T$, where $\lambda$ is the spherical longitude, $\varphi$ is the spherical latitude, $\boldsymbol{x}_k = R \cdot \boldsymbol{r}_k$, $P_n$ is the Legendre polynomial of degree $n$ and $B_n$ is a Legendre coefficient which specifies the shape of the SRBF.

With the spherical basis function (Eq. 1), a harmonic signal $F(\boldsymbol{x})$ can be described as

$$F(\boldsymbol{x}) = \sum_{k=1}^{K} d_k B(\boldsymbol{x}, \boldsymbol{x}_k) \, , \tag{2}$$

where $K$ is the number of basis functions. The unknown coefficients $d_k$ can be evaluated from the observations. As will be shown in the following subsection, using these coefficients, any functional of $F(\boldsymbol{x})$ can be described.

### 2.1 Gravitational functionals

Various functionals can be derived from the gravitational potential based on field transformations:



**Disturbing potential**

The disturbing potential $T$ is defined as the difference between the total gravity potential $W$ and the normal gravity potential $U$

$$T = W - U , \tag{3}$$

where the latter is the potential related to the level ellipsoid. The gravity potential $W$ consists of two parts, the gravitational
potential $V$ and the centrifugal potential $Z$, i.e.

$$W = V + Z . \tag{4}$$

Combining Eq. (3) and Eq. (4) yields

$$T = V - U + Z \tag{5}$$

(Hofmann-Wellenhof and Moritz, 2005).

**Gravitational potential difference**

The satellite gravity field mission Gravity Recovery and Climate Experiment (GRACE, Tapley et al., 2004) consists of two
satellites A and B. An observation $\Delta V_{AB}$ can be interpreted as the difference between the gravitational potential values $V$ of A
and B, i.e. $\Delta V_{AB} = V(\boldsymbol{x}^A) - V(\boldsymbol{x}^B)$. Including the measurement error $e$, the observation equation reads

$$\Delta V(\boldsymbol{x}^A, \boldsymbol{x}^B) + e(\boldsymbol{x}^A, \boldsymbol{x}^B) = V(\boldsymbol{x}^A) - V(\boldsymbol{x}^B) + e(\boldsymbol{x}^A, \boldsymbol{x}^B) = \sum_{k=1}^{K} d_k B(\boldsymbol{x}^A, \boldsymbol{x}^B, \boldsymbol{x}_k) ; \tag{6}$$

the function $B(\boldsymbol{x}^A, \boldsymbol{x}^B, \boldsymbol{x}_k)$ is given in Table 1.

**Gravity disturbance**

The gravity disturbance is generally used in airborne and terrestrial gravity field determination (Alberts, 2009). The gravity
disturbance vector $\delta\boldsymbol{g}$ is expressed as the gradient of the disturbing potential $T$

$$\delta\boldsymbol{g} = \left[\frac{\partial T}{\partial x}, \frac{\partial T}{\partial y}, \frac{\partial T}{\partial z}\right]^{\mathrm{T}} = \mathrm{grad} T . \tag{7}$$

In spherical approximation, the magnitude of the gravity disturbance can be written as

$$\delta g = -\frac{\partial T}{\partial r} = -T_r , \tag{8}$$

its observation equation reads

$$\delta g(\boldsymbol{x}) + e(\boldsymbol{x}) = \sum_{k=1}^{K} d_k B_r(\boldsymbol{x}, \boldsymbol{x}_k) , \tag{9}$$

where the basis function $B_r(\boldsymbol{x}, \boldsymbol{x}_k)$ is given in Table 1.





**Gravity gradients**

Equipped with a 3-axis gradiometer, the satellite mission Gravity Field and Steady-State Ocean Circulation Explorer (GOCE, Rummel et al., 2002) observed the tensor $\Delta \boldsymbol{V}$ of the gravity gradients $V_{ab}$ with $a, b \in \{x, y, z\}$, i.e. all second-order derivatives of the gravitational potential $V$

$$\Delta \boldsymbol{V} = \begin{bmatrix} V_{xx} & V_{xy} & V_{xz} \\ V_{yx} & V_{yy} & V_{yz} \\ V_{zx} & V_{zy} & V_{zz} \end{bmatrix} \tag{10}$$

with $V_{xy} = V_{yx}$ , $V_{xz} = V_{zx}$ , $V_{yz} = V_{zy}$ and trace $(\Delta \boldsymbol{V}) = 0$ due to the Laplacian differential equation. In spherical approximation $V_{zz} \approx V_{rr} = \frac{\partial^2 V}{\partial r^2}$ holds and the observation equation reads

$$V_{rr}(\boldsymbol{x}) + e(\boldsymbol{x}) = \sum_{k=1}^{K} d_k B_{rr}(\boldsymbol{x}, \boldsymbol{x}_k) ; \tag{11}$$

the basis function $B_{rr}(\boldsymbol{x}, \boldsymbol{x}_k)$ can be found in Table 1.

## 2.2 Gravitational functionals in terms of SRBFs

In this study, all observations are simulated in the sense of disturbing gravity field quantities, i.e. disturbing potential differences $\Delta T = T(\boldsymbol{x}^A) - T(\boldsymbol{x}^B)$ for GRACE, the first order radial derivatives $T_r$ for terrestrial and airborne observations as well as the second order radial derivatives $T_{rr}$ for GOCE. For each type of observation, the adapted basis functions are listed in Table 1.

Basis functions adapted to other functionals of the disturbing potential which are not used here are listed in Koop (1993), Lieb et al. (2016) and Lieb (2017).

## 2.3 Types of spherical radial basis functions

Since it is not possible to reach perfect localization in both the spectral and spatial domain due to the uncertainty principle (Freeden, 1998; Ozawa, 2003), we want to find an appropriate compromise between these two domains. Different types of SRBFs can be found amongst others in Schmidt (2007), Bentel (2013a, b). Three types of SRBFs are studied in this work, including functions with smoothing features (Blackman and Cubic Polynomial) and without smoothing features (Shannon). The Shannon function has the simplest representation; its Legendre coefficients are given by

$$B_n = \begin{cases} 1 & \text{for } n \in [0, N_{max}] \\ 0 & \text{else} \end{cases} \tag{12}$$

The Blackman function is derived from the Blackman window; its Legendre coefficients are given by

$$B_n = \begin{cases} 1 & \text{for } n \in [0, n_1) \\ (A(n))^2 & \text{for } n \in [n_1, N_{max}] \\ 0 & \text{else} \end{cases} \tag{13}$$

where

$$A(n) = \frac{21}{50} - \frac{1}{2}\cos\left(\frac{2\pi(n - N_{max})}{2(N_{max} - n_1)}\right) + \frac{2}{25}\cos\left(\frac{4\pi(n - N_{max})}{2(N_{max} - n_1)}\right) \tag{14}$$





In case of the Cubic Polynomial (CuP) function, the Legendre coefficients are given by a cubic polynomial, namely

$$B_n = \begin{cases} \left(1 - \frac{n}{N_{max}}\right)^2 \left(1 + \frac{2n}{N_{max}}\right) & \text{for } n \in [0, N_{max}] \\ 0 & \text{else} \end{cases} \tag{15}$$

$N_{max}$ is a certain degree to which the SRBFs are expanded, representing the cut-off degree in the frequency domain. These three functions and their referring Legendre coefficients for $N_{max} = 255$ are plotted in Figure 1, it visualizes the characteristics

in the spatial and the spectral domain correspondingly.

In the spatial domain, the Shannon function has the sharpest peak but the strongest oscillations; the Blackman function has less oscillations than the Shannon and the CuP function has the weakest ones. However, in the spectral domain, the Shannon function gets the strongest localization due to its exact band limitation without losing any spectral information; the Blackman

function has a smoothing decay at the higher frequencies of the function; the CuP function has an even stronger smoothing decay and thus, extracts less spectral information compared to Shannon and Blackman. Therefore, in this study, we use the Shannon function for estimating the coefficients $d_k$ within the analysis step to reduce the loss of signal content, and then use the Blackman function for the synthesis step to reduce erroneous systematic effects due to oscillations. The same experiments will be applied using the CuP function as well to test if different SRBFs will affect the performance of the regularization

method.

## 3. Parameter Estimation

To determine the unknown coefficients $d_k$ from Eq. (2), the method of parameter estimation is used in this study. This process allows different types of observations to be combined considering their individual strength and favorable features (Schmidt et al., 2015).

### 3.1 Gauss-Markov Model

For one single observation, i.e. a functional of the disturbing potential $T$, the observation equation reads

$$F(\pmb{x}) + e(\pmb{x}) = \sum_{k=1}^{K} d_k B(\pmb{x}, \pmb{x}_k) , \tag{16}$$

$B(\pmb{x}, \pmb{x}_k)$ represents the adapted SRBFs as listed in Table 1. Collecting the observations $F(\pmb{x}_1), F(\pmb{x}_2), \ldots, F(\pmb{x}_n)$ in the $n \times 1$ observation vector $\pmb{f}$, the Gauss-Markov model

$$\pmb{f} + \pmb{e} = \pmb{A}\pmb{d} \qquad \text{with} \qquad D(\pmb{f}) = \sigma^2 \pmb{P}^{-1} \tag{17}$$

(deterministic part)          (stochastic part)

can be set up. In the deterministic part, $\pmb{e} = [e(\pmb{x}_1), e(\pmb{x}_2), \ldots, e(\pmb{x}_n)]^T$ is the $n \times 1$ vector of the observation errors and $\pmb{A} = [B(\pmb{x}, \pmb{x}_k)]$ is the $n \times K$ design matrix containing the corresponding basis functions. In the stochastic part, $D(\pmb{f})$ is the $n \times n$ covariance matrix of the observation vector $\pmb{f}$ with $\sigma^2$ the unknown variance factor and $\pmb{P}$ the given positive definite weight

matrix.



The least-squares adjustment can be applied to the model (Eq. 17) as long as the design matrix is of full column rank (Schmidt et al., 2015). Then the solution reads

$$\widehat{\boldsymbol{d}} = (\boldsymbol{A}^T \boldsymbol{P} \boldsymbol{A})^{-1} \boldsymbol{A}^T \boldsymbol{P} \boldsymbol{f} \qquad (18)$$

$$D(\widehat{\boldsymbol{d}}) = \sigma^2 (\boldsymbol{A}^T \boldsymbol{P} \boldsymbol{A})^{-1} \ . \qquad (19)$$

Due to the aforementioned three reasons, the normal equation matrix $\boldsymbol{N} = \boldsymbol{A}^T \boldsymbol{P} \boldsymbol{A}$ is ill-posed or even singular. For handling this problem, we introduce an additional linear model

$$\boldsymbol{\mu}_d + \boldsymbol{e}_d = \boldsymbol{d} \quad \text{with} \quad D(\boldsymbol{\mu}_d) = \sigma_d^2 \boldsymbol{P}_d^{-1} \qquad (20)$$

as prior information. $\boldsymbol{\mu}_d$ is the $K \times 1$ expectation vector of the coefficient vector $\boldsymbol{d}$, $\boldsymbol{e}_d$ is the corresponding error vector and $D(\boldsymbol{\mu}_d)$ is the $K \times K$ covariance matrix of the prior information with $\sigma_d^2$ the unknown variance factor and $\boldsymbol{P}_d$ the given positive

definite weight matrix.

Combining the two models (Eq. 17) and (Eq. 20) yields the extended linear model

$$\begin{bmatrix} \boldsymbol{f} \\ \boldsymbol{\mu}_d \end{bmatrix} + \begin{bmatrix} \boldsymbol{e} \\ \boldsymbol{e}_d \end{bmatrix} = \begin{bmatrix} \boldsymbol{A} \\ \boldsymbol{I} \end{bmatrix} \boldsymbol{d} \text{ with } D\left( \begin{bmatrix} \boldsymbol{f} \\ \boldsymbol{\mu}_d \end{bmatrix} \right) = \sigma^2 \begin{bmatrix} \boldsymbol{P}^{-1} & \boldsymbol{0} \\ \boldsymbol{0} & \boldsymbol{0} \end{bmatrix} + \sigma_d^2 \begin{bmatrix} \boldsymbol{0} & \boldsymbol{0} \\ \boldsymbol{0} & \boldsymbol{P}_d^{-1} \end{bmatrix} \qquad (21)$$

Now the least-squares adjustment can be applied and leads to the normal equations

$$\left( \frac{1}{\sigma^2} \boldsymbol{A}^T \boldsymbol{P} \boldsymbol{A} + \frac{1}{\sigma_d^2} \boldsymbol{P}_d \right) \widehat{\boldsymbol{d}} = \frac{1}{\sigma^2} \boldsymbol{A}^T \boldsymbol{P} \boldsymbol{f} + \frac{1}{\sigma_d^2} \boldsymbol{P}_d \boldsymbol{\mu}_d \ . \qquad (22)$$

The variance factors $\sigma^2$ and $\sigma_d^2$ can either be given as prior information or estimated within a VCE, then the solution reads

$$\widehat{\boldsymbol{d}} = (\boldsymbol{A}^T \boldsymbol{P} \boldsymbol{A} + \lambda \boldsymbol{P}_d)^{-1} (\boldsymbol{A}^T \boldsymbol{P} \boldsymbol{f} + \lambda \boldsymbol{P}_d \boldsymbol{\mu}_d) \qquad (23)$$

$$D(\widehat{\boldsymbol{d}}) = \sigma^2 (\boldsymbol{A}^T \boldsymbol{P} \boldsymbol{A} + \lambda \boldsymbol{P}_d)^{-1} \ , \qquad (24)$$

wherein $\lambda = \sigma^2 / \sigma_d^2$ is the regularization parameter, see Koch and Kusche (2002) and Schmidt et al. (2007).

### 3.2 Combination models

To combine different types of heterogeneous data sets for regional gravity field modeling, combination models (CMs) need to be set up (see e.g. Schmidt et al., 2015).

In general, let $\boldsymbol{f}_p$ with $p = 1, \dots, P$ be the observation vector of the $p^{th}$ observation technique, such as $\boldsymbol{f}_p = \left[ F_p(\boldsymbol{x}_1), F_p(\boldsymbol{x}_2), \dots, F_p\left( \boldsymbol{x}_{n_p} \right) \right]^T$, $\boldsymbol{e}_p$ and $\boldsymbol{A}_p$ are the corresponding error vector and the design matrix. Note that for different techniques, the data are observed as different gravitational functionals and thus, the adapted SRBFs as discussed in the Sect.

2.1 should be applied accordingly, and $\boldsymbol{A}_p = [B_p(\boldsymbol{x}, \boldsymbol{x}_k)]$.

For the combination of the $P$ observation techniques, including the additional linear model for the prior information (Eq. 20), an extended Gauss-Markov model can be formulated (Lieb, 2017)

$$\begin{bmatrix} \boldsymbol{f}_1 \\ \boldsymbol{f}_2 \\ \vdots \\ \boldsymbol{f}_p \\ \boldsymbol{\mu}_d \end{bmatrix} + \begin{bmatrix} \boldsymbol{e}_1 \\ \boldsymbol{e}_2 \\ \vdots \\ \boldsymbol{e}_p \\ \boldsymbol{e}_d \end{bmatrix} = \begin{bmatrix} \boldsymbol{A}_1 \\ \boldsymbol{A}_2 \\ \vdots \\ \boldsymbol{A}_p \\ \boldsymbol{I} \end{bmatrix} \cdot \boldsymbol{d} \text{ with } D\left( \begin{bmatrix} \boldsymbol{f}_1 \\ \boldsymbol{f}_2 \\ \vdots \\ \boldsymbol{f}_p \\ \boldsymbol{\mu}_d \end{bmatrix} \right) = \begin{bmatrix} \sigma_1^2 \boldsymbol{P}_1^{-1} & \boldsymbol{0} & \boldsymbol{0} & \cdots & \boldsymbol{0} \\ \boldsymbol{0} & \sigma_1^2 \boldsymbol{P}_1^{-1} & \vdots & \vdots & \vdots \\ \vdots & \boldsymbol{0} & \ddots & \sigma_1^2 \boldsymbol{P}_1^{-1} & \vdots \\ \vdots & \vdots & \vdots & & \boldsymbol{0} \\ \boldsymbol{0} & \boldsymbol{0} & \cdots & \boldsymbol{0} & \sigma_1^2 \boldsymbol{P}_1^{-1} \end{bmatrix}, \qquad (25)$$





where $\boldsymbol{P}_p$ is the $n_p \times n_p$ positive definite weight matrix of the $p^{th}$ observation technique.

**CM 1**: We assume that for each technique $p$ the variance factor $\sigma_p^2$ is the same, i.e. $\sigma_1^2 = \sigma_2^2 = \cdots = \sigma_p^2 = \sigma^2$. Hence, with $\boldsymbol{f} = \left[\boldsymbol{f}_1^T, \boldsymbol{f}_2^T, \dots, \boldsymbol{f}_p^T\right]^T$ the covariance matrix becomes

$$5 \quad \mathrm{D}\left(\begin{bmatrix} \boldsymbol{f} \\ \boldsymbol{\mu}_d \end{bmatrix}\right) = \begin{bmatrix} \sigma^2 \boldsymbol{P}^{-1} & \boldsymbol{0} \\ \boldsymbol{0} & \sigma_d^2 \boldsymbol{P}_d^{-1} \end{bmatrix}. \tag{26}$$

Thus, this combination model is transferred into the single observation model (Eq. 21) and the estimation of the coefficient vector $\hat{\boldsymbol{d}}$ can be obtained from the Eq. (23) if the regularization parameter $\lambda$ is known.

**CM 2**: Since different data sets have different spatial resolution and spectral characteristics, the assumption made in CM 1 is not always the most accurate case. Therefore, they can be combined in a way which takes the individual variance component of each observation technique into account.

Applying the least-squares method to Eq. (25), the extended normal equations read

$$\left(\sum_{p=1}^{P} \left(\frac{1}{\sigma_p^2} \boldsymbol{A}_p^T \boldsymbol{P}_p \boldsymbol{A}_p\right) + \frac{1}{\sigma_d^2} \boldsymbol{P}_d\right) \hat{\boldsymbol{d}} = \sum_{p=1}^{P} \left(\frac{1}{\sigma_p^2} \boldsymbol{A}_p^T \boldsymbol{P}_p \boldsymbol{f}_p\right) + \frac{1}{\sigma_d^2} \boldsymbol{P}_d \boldsymbol{\mu}_d. \tag{27}$$

Solving Eq. (27) with given values for the variance factors, we obtain

$$15 \quad \hat{\boldsymbol{d}} = \left(\sum_{p=1}^{P} \left(\frac{1}{\sigma_p^2} \boldsymbol{A}_p^T \boldsymbol{P}_p \boldsymbol{A}_p\right) + \frac{1}{\sigma_d^2} \boldsymbol{P}_d\right)^{-1} \left(\sum_{p=1}^{P} \left(\frac{1}{\sigma_p^2} \boldsymbol{A}_p^T \boldsymbol{P}_p \boldsymbol{f}_p\right) + \frac{1}{\sigma_d^2} \boldsymbol{P}_d \boldsymbol{\mu}_d\right) \tag{28}$$

with the covariance matrix

$$D(\hat{\boldsymbol{d}}) = \left(\sum_{p=1}^{P} \left(\frac{1}{\sigma_p^2} \boldsymbol{A}_p^T \boldsymbol{P}_p \boldsymbol{A}_p\right) + \frac{1}{\sigma_d^2} \boldsymbol{P}_d\right)^{-1} \tag{29}$$

Equation (28) can be rewritten as

$$\hat{\boldsymbol{d}} = \left(\sum_{p=1}^{P} \left(\omega_p \boldsymbol{A}_p^T \boldsymbol{P}_p \boldsymbol{A}_p\right) + \lambda \boldsymbol{P}_d\right)^{-1} \left(\sum_{p=1}^{P} \left(\omega_p \boldsymbol{A}_p^T \boldsymbol{P}_p \boldsymbol{f}_p\right) + \lambda \boldsymbol{P}_d \boldsymbol{\mu}_d\right), \tag{30}$$

20 such that the regularization parameter $\lambda = \sigma_1^2 / \sigma_d^2$ (Koch and Kusche, 2002), and the factors $\omega_1 = \sigma_1^2 / \sigma_1^2 = 1$, $\omega_2 = \sigma_1^2 / \sigma_2^2, \dots, \omega_p = \sigma_1^2 / \sigma_p^2$ express the relative weightings of the observation vector $\boldsymbol{f}_p$ with respect to $\boldsymbol{f}_1$.

## 4. Choice of the regularization parameter

A critical question of regularization is the selection of an appropriate regularization parameter $\lambda$ (Kusche and Klees, 2002). In the following, the L-curve method and the VCE will be explained in more detail. Finally, two new proposed methods are presented as a combination of VCE and the L-curve method.





### 4.1 L-curve method

The L-curve is a graphical procedure for regularization (Hansen, 1990; Hansen and OLeaary, 1993; Bouman, 1998; Hansen, 2000). Plotting the norm of the regularized solution $\left\| \widehat{\boldsymbol{d}}_\lambda - \boldsymbol{\mu}_d \right\|$ against the norm of the residuals $\|\widehat{\boldsymbol{e}}\| = \left\| \boldsymbol{A}\widehat{\boldsymbol{d}}_\lambda - \boldsymbol{f} \right\|$ by changing the numerical value for the regularization parameter $\lambda$ shows a typical 'L-curve' behavior, i.e. it looks like the capital

letter 'L' (see Fig. 3). The corner point in this 'L-shaped' curve means a compromise of the minimization of the solution norm and the residual norm, and thus can be interpreted as the 'best fit' point that corresponds to the desired regularization parameter.

### 4.2 VCE

Variance component estimation is a useful method when several data sets need to be combined in a parameter estimation procedure (Koch, 1999; Naeimi, 2015). The variance components are estimated by an iterative process, starting from initial

values for $\sigma_p^2, \sigma_d^2$ and ending in the convergence point. The estimations read

$$\begin{cases} \widehat{\sigma}_p^2 = \dfrac{\widehat{\boldsymbol{e}}_p^T \boldsymbol{P}_p \widehat{\boldsymbol{e}}_p}{r_p} \\ \widehat{\sigma}_d^2 = \dfrac{\widehat{\boldsymbol{e}}_d^T \boldsymbol{P}_d \widehat{\boldsymbol{e}}_d}{r_d} \end{cases} \tag{31}$$

where $r_p$ and $r_d$ are the partial redundancies computed following Koch and Kusche (2002), and the residual vectors $\widehat{\boldsymbol{e}}_p$ and $\widehat{\boldsymbol{e}}_d$ are given by

$$\begin{cases} \widehat{\boldsymbol{e}}_p = \boldsymbol{A}_p \widehat{\boldsymbol{d}} - \boldsymbol{f}_p \\ \ \widehat{\boldsymbol{e}}_d = \widehat{\boldsymbol{d}} - \boldsymbol{\mu}_d \end{cases} . \tag{32}$$

### 4.3 Combination of VCE and the L-curve method

Two ways of combining VCE and the L-Curve method are discussed and applied in this study, namely 'VCE + L-curve method' and 'L-curve method + VCE'.

#### 1. 'VCE + L-curve method'

Figure 2 illustrates the procedure of the 'VCE + L-curve method'. In the first step, the VCE is applied according to the

combination model CM 2. This step gives the regularization parameter $\lambda_{\text{VCE}}$ and the relative weighting factors $\omega_p$. The weighting factors $\omega_p$ are then used in the L-curve method to regenerate a new regularization parameter $\lambda$ (Fig. 3). In this case, the coefficient vector $\widehat{\boldsymbol{d}} = \widehat{\boldsymbol{d}}_\lambda$ is calculated for a group of changing regularization parameters $\lambda$ using Eq. (30).

Thus, the final solution is computed using Eq. (30) with the weights $\omega_p$ and the new regularization parameter $\lambda$ from the L-curve criterion.





## 2. 'L-curve method + VCE'

Figure 4 illustrates the procedure of the 'L-curve method + VCE'. In opposite to the 'VCE + L-curve method', in the 'L-curve method + VCE' the L-curve method is applied according to the combination model CM 1 first. A regularization parameter $\lambda_{L-curve}$ is obtained in the first step, and it is used for defining the value of $\sigma_d^2$ in the variance component estimation.

In the second step, the VCE is applied according to CM 2, with initial values $\sigma_1^2 = \sigma_2^2 = \cdots = \sigma_p^2 = 1$ and $\sigma_d^2 = \sigma_1^2/\lambda_{L-curve}$. After each iteration within the VCE, the value of $\sigma_d^2$ is set to $\sigma_1^2/\lambda_{L-curve}$ again, with the new value of $\sigma_1^2$ obtained in this iteration. In this case, the regularization parameter $\lambda$ calculated from the L-curve method will be kept, but the relative weighting factors $\omega_p$ are recomputed in each iteration step. The final solution is computed using Eq. (30) with the relative weights $\omega_p$ and the regularization parameter $\lambda_{L-curve}$.

To summarize, the purpose of these two proposed methods is to bring the L-curve method and VCE together, and test if the regularization results will be improved. 'VCE + L-curve method' fixes the relative weights of each observation technique first and tries to find a 'best fit' regularization parameter; while 'L-curve + VCE' fixes the regularization parameter first and then tries to find the relative weights for each observation technique.

## 5. Numerical investigation

### 5.1 Data description

The data used in this study are obtained from the ICCT (Inter-Commission Committee on Theory) Joint Study Group 0.3, part of the IAG (International Association of Geodesy) programme running from 2011 to 2015. The observation data are simulated from the Earth gravitational model EGM2008 (Pavlis et al., 2012) and are provided along with simulated observation noise. The standard deviation of the white noise is set to $8 \cdot 10^{-4}$ m$^2$/s$^2$ for GRACE, 10 mE for GOCE, 10 μGal for the terrestrial

data and 1 mGal for the airborne data. The study area chosen here is 'Europe' where the validation data are also provided on geographic grid points in terms of disturbing potential values $T$ with different grid resolutions (30'×30' and 5'×5') and different spherical harmonic resolutions (maximum degree 250 and 2190).

Figure 5 illustrates the available observation data as well as the validation data. Three types of observations are included:

1.  Satellite data: provided along the real satellite orbits of GRACE (green tracks in Fig. 5) and GOCE (red tracks). GRACE

data span a one month period, and GOCE data cover a full repeat cycle of 61 days.

2.  Terrestrial data: provided on a regular grid with two different resolutions, one over an area of 20º × 30º (latitude × longitude) with a grid spacing of 30' (blue shaded area) and the other one over an inner area of 6º × 10º with a grid spacing of 5' (yellow shaded area).

3.  Airborne data: provided on two different flight tracks, one over the Adriatic Sea (magenta shaded area) and the other one

over Corsica connecting Southern Europe with Northern Africa (cyan shaded area).



## 5.2 Model configuration

A Remove-Compute-Restore approach is applied in this study, i.e., from each type of observations, the background model EGM96 (Lemoine et al., 1998) up to spherical harmonic degree 60 is removed and restored after computation. The background model serves additionally as prior information, and in this case, the expectation vector $\boldsymbol{\mu}_d$ can be assumed to be the zero vector

(Lieb, 2017). We assume that the coefficients have the same accuracies and are uncorrelated, thus, $\boldsymbol{P}_d = \boldsymbol{I}$, where $\boldsymbol{I}$ denotes the identity matrix. Further, we set $\boldsymbol{P}_p = \boldsymbol{I}$ by assuming the measurement errors to be uncorrelated and the observations to have the same accuracy.

In the analysis step we use the Shannon function for estimating the vector $\widehat{\boldsymbol{d}}$ of the unknown coefficients $d_k$ related to the grid points $P_k$ within the area $\partial\Omega_C$ of computation (see Fig. 6) from the measurements available within the area $\partial\Omega_O$ of

observations. In the following synthesis step the Blackman function is used for calculating the output gravity functional within the area $\partial\Omega_I$ of investigation. It has to be mentioned that the points $P_k$ within the area $\partial\Omega_C$ of computation are defined by a Reuter grid.

Margins $\eta$ have to be defined between the three areas to minimize edge effects in the computation process (Lieb et al., 2016). In this study, we conducted the experiments using different margin sizes, and the one which gives the smallest RMS error is

finally chosen.

The aforementioned five methods for choosing the regularization parameter (Table 2) are applied to six groups of data sets (Table 3), respectively.

The computed disturbing potential $T_c$ is compared with the corresponding validation data $T_v$ and assessed following three criteria:

1.   Root mean square error (RMS) of the computed disturbing potential $T_c$ with respect to the validation data $T_v$ over the investigation area $\partial\Omega_I$

$$\text{RMS} = \sqrt{\frac{\sum_{n_{\text{points}}} (T_v - T_c)^2}{n_{\text{points}}}} \ . \tag{33}$$

2.   Correlation coefficient between the estimated coefficients $d_k$ collected in the vector $\widehat{\boldsymbol{d}}$ and the validation data $T_v$. The reason that this correlation can be used as a criterion is that the estimated coefficients $d_k$ reflect the energy of the

gravity field at their locations. The energy $E_k$ at location $\boldsymbol{x}_k$ can be expressed by

$$E_k = d_k \sum_{p=1}^{K} d_p \sum_{n=0}^{N_{max}} \frac{2n+1}{4\pi} B_n^2 P_n(\boldsymbol{r}_p^{\text{T}} \boldsymbol{r}_k) \ , \tag{34}$$

(Lieb, 2017). For $N_{max} \to \infty$ and $B_n = 1$, the relation (Eq. 34) equals approximately $E_k = d_k^2$.

The same criterion is used as a quality measure by Bentel et al. (2013a) and Naeimi et al. (2015).

3.   Correlation coefficient between the recovered gravity field $T_c$ and the validation data $T_v$.



### 5.3 Results

For the sake of brevity, only the results of two study cases (A and F) are detailed here. However, results obtained from all study cases, including the RMS error and the correlations between the estimated coefficients $d_k$ and the validation data $T_v$ of each method are summarized in the Tables 6 and 7, respectively. To test the stability of our new methods, the same experiments

are also applied using the CuP function for analysis and synthesis as a comparison scenario. The results are listed in the Tables 8 and 9.

### Study case A

GRACE observations and terrestrial observations with a 30' resolution are combined. The maximum degree in the expansion in terms of SRBF is set to $N_{max} = 300$ for the Shannon function (Eq. 12) in the analysis step as well as

$n_1 = 250$ and $N_{max} = 350$ for the Blackman function (Eq. 13) in the synthesis step. The margin $\eta$ between the different areas (Fig. 6) is chosen to 4º. Five solutions are performed according to the five parameter choice methods 1) to 5) listed in Table 2. For each solution, the estimated coefficients $d_k$, the calculated disturbing potential $T_c$ as well as its difference to the validation data are plotted in Fig. 7. The results for the three criteria measures from above are listed in Table 4.

The correlations between the modelled gravity field $T_c$ and the validation data $T_v$ for all parameter choice methods are rather

satisfying. However, the CM 1 based methods give slightly smaller correlations than the others. The last three methods are comparable and provide better results than the CM 1 based methods with respect to both RMS values and correlations. The lowest RMS error is obtained from the 'VCE + L-curve method' which is 3.61 m²/s². This method also delivers the highest correlation between the estimated coefficient $d_k$ and the validation data. "L-curve method + VCE" gives the second-best RMS value which is 3.64 m²/s², followed by 'VCE based on CM 2'. The same rank applies to the correlation between the estimated

coefficient $d_k$ and the validation data. The largest RMS value and smallest correlation are both obtained from parameter choice methods based on CM 1. Among the methods based on CM 1, the L-curve method provides smaller RMS value compare to VCE, but smaller correlation factor as well.

It is worth clarifying that the solution obtained from the 'L-curve method + VCE' is not unique. Due to the fact that the regularization parameter $\lambda_{L-curve}$ is fixed during VCE, the results change when it refers to different observation techniques.

Here, two solutions are obtained by setting GRACE data and the terrestrial data as the reference observation technique, respectively. In this study case, the better result (smaller RMS value and larger correlation) is the one when choosing the terrestrial data as the reference. Generally, a better solution is obtained when the terrestrial data are chosen as the reference, and the results for 'L-curve method + VCE' listed in this paper are always the better ones.

### Study case F

In case F, the maximum degree in the expansion in terms of SRBF is set to $N_{max} = 1050$ for the Shannon function (Eq. 12) in the analysis step as well as $n_1 = 900$ and $N_{max} = 1100$ for the Blackman function (Eq. 13) in the synthesis step. The



margin $\eta$ between the different areas (Fig. 6) is chosen to 2°. For each method, the estimated coefficients $d_k$, the calculated disturbing potential $T_c$ as well as its difference to the validation data are plotted in Fig. 8. The results for the three criteria measures from above are listed in Table 5.

Compared to the study case A, the results in the study case F shows a general improvement, in terms of all three criteria. The
correlations between the estimated coefficients and the validation data are promising, which can be observed in Fig. 8. It shows that the estimated coefficients reflect the energy of the recovered gravity signal rather well. 'VCE + L-curve method' still provides the smallest RMS error 0.78 m²/s² as well as the highest correlations. 'L-curve method + VCE' gives RMS value 0.80 m²/s², followed by 'VCE based on CM 2' with 0.82 m²/s². The same rank applies to the correlation between the estimated coefficient $d_k$ and the validation data.

The largest RMS values and the smallest correlations are still both obtained from parameter choice methods based on CM 1. The differences between CM 1 and CM 2 based methods are considerable. It could be caused by the large variation between the spectral resolution of GRACE, GOCE, terrestrial and airborne data. Therefore, giving each observation technique a relative weight in the combination might help to provide better results. Among the two methods based on CM 1, the L-curve method provides worse RMS value in comparison to VCE but, a much more significant correlation between $d_k$ and $T_v$.

**Results of all study cases**

Table 6 lists the RMS values of each method obtained from all the study cases. Table 7 lists the correlations between the estimated coefficients $d_k$ and the validation data $T_v$ of each method obtained from all the study cases. The best results (smallest RMS value and largest correlation) in each study case are bold-typed and the second bests are italicized.

Based on these results, the following conclusions can be drawn:

1. From the five parameter choice methods considered here, 'VCE + L-curve method' performs the best, and always gives the smallest RMS error as well as the largest correlation.

2. 'L-curve method + VCE' and 'VCE based on CM 2' also show a good and stable performance, especially when the spectral resolution of the combined data sets differs. 'L-curve method + VCE' generally outperforms 'VCE based on CM 2' slightly.

3. The results in terms of RMS value and correlation are consistent, i.e., in most cases, the regularization parameter choice method which gives the smaller RMS error also delivers the larger correlation.

4. Generally, results provided by CM 1 based methods are not as good as the others. The larger the spectral resolution between each observation technique is, the worse these methods perform (e.g. case E and F). However, for case D where the combined data sets have similar resolution, 'the L-curve method based on CM 1' perform even better than

'L-curve method + VCE' and 'VCE based on CM 2'.

5. The results also indicate that when the Shannon function is used for analysis and the same combination model CM 1 is used, the L-curve method generally outperforms VCE (case A, B, C, D, E regarding the RMS value and case B, C, D, E, F regarding the correlation).



**Results using the CuP function**

The Tables 8 and 9 list the RMS values as well as the correlations between the estimated coefficients $d_k$ and the validation data $T_v$ of each method when the CuP function is used. Again, the best results (smallest RMS value and largest correlation) in each study case are bold-typed and the second bests are italicized.

In this comparison scenario, 'VCE + L-curve method' still delivers always the best results for all six study cases in terms of both RMS value and correlation, which proves that its performance does not depend on the type of SRBF used. The performance of 'L-curve method + VCE' and 'VCE based on CM 2' are also stable and provide rather good results. The RMS values provided by 'the L-curve method based on CM 1' are not as good as those obtained when using the Shannon function for analysis. While in opposite, the correlations provided by 'the L-curve method based on CM 1' are better than those obtained

when using the Shannon function. This behavior is consistent with the publication of Naeimi (2015).

It was not the purpose of this paper to compare the performance of different types of SRBFs. However, it can be observed clearly from the Tables 6 and 7 as well as 8 and 9 that when the CuP function is used, the correlations between the estimated coefficients $d_k$ and the validation data $T_v$ dropped significantly; especially, for the study cases D, E and F, where the maximum degree $N_{max}$ of the expansion in terms of SRBFs is high ($N_{max} = 1100$). This can probably be explained by the fact that

smoothing functions lead to the loss of some signal components, particularly the higher frequencies. The results regarding the RMS value when using the Shannon/Blackman functions are similar to those when the CuP/CuP functions are used. For study cases C, D, E and F, the RMS error are slightly smaller when the CuP/CuP functions are applied. It is worth mentioning that the same experiments were also implemented using the Shannon function for analysis and the CuP function for synthesis. The performance of each method stays the same and 'VCE + L-curve method' still delivers the best results. However, the RMS

values of each study case are generally larger in comparison to the results when using Shannon function for analysis but Blackman for synthesis. Thus, the detailed results of that application are not listed here due to the length of this paper.

**6. Summary and conclusions**

We discussed the parameter estimation using SRBFs for combining heterogeneous data sets, and two types of combination models were introduced. CM 1 merges all types of observations into one observation vector without weighting, and CM 2

gives each type of observation techniques a relative weight using VCE. Based on these two combination models, the determination of the regularization parameter is investigated using simulated satellite, terrestrial and airborne data.

We presented five methods for choosing the regularization parameter, including 'the L-curve method based on CM 1'; 'VCE based on CM 1'; 'VCE based on CM 2' and two newly proposed methods which are 'VCE + L-curve method' and 'L-curve method + VCE'. Each method is applied to six groups of data sets, and the results are compared to the validation data with

corresponding spatial and spectral resolutions. The investigation showed that our new proposed 'VCE + L-curve method' gives the best results in all the six study cases; 'L-curve method + VCE' also provides fairly good solutions. The results also indicate




that the larger the spectral resolution differs between each type of observation technique, the better 'VCE based on CM 2' and 'L-curve method + VCE' perform compared to the L-curve method or VCE based on CM 1; which is reasonable.

We also carried out the same investigation using the CuP function for comparison scenario to test the dependency of our new methods against the type of SRBFs used. In this scenario, the performance of the L-curve method is slightly lower than the

one obtained using the Shannon function. This behavior is consistent with the results from the literature. Our newly proposed 'VCE + L-curve method' still provides the best results in all the six cases and the 'L-curve method + VCE' is also performing well. Thus, we are able to conclude that our new methods provide good regularization results for different observation combinations and are stable regardless of the type of SRBF used. From our investigation, we conclude that 'VCE + L-curve method' is the best choice among those five methods for the determination of the regularization parameter.

In future, a primary concern is to apply the newly devised methods using more types of SRBFs, so that the performance of different SRBFs can be compared while making sure that the differences in results are not coming from the regularization method. In addition, further investigations are planned for using real observations instead of simulated data sets.

*Competing interests.* The authors declare that they have no competing interests.

*Acknowledgements.* This study was conducted in the framework of the project 'Optimally combined regional geoid models for the realization of height systems in developing countries', the authors would like to thank the German Research Foundation (DFG) for funding this project. We also acknowledge the ICCT Joint Study Group 0.3 for providing the data sets. We are grateful to Katrin Bentel for kindly providing support and the basic programme related to this work. Furthermore, the authors

acknowledge the developers of the Generic Mapping Tool (GMT) mainly used for generating the figures in this work.

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



**Table 1: Adapted basis functions for each type of observation**

| observation | adapted basis function $B$ |
|---|---|
| GRACE | $B(\boldsymbol{x}^A, \boldsymbol{x}^B, \boldsymbol{x}_k) = \sum_{n=0}^{\infty} \frac{2n+1}{4\pi} B_n \left\{ \left(\frac{R}{r^A}\right)^{n+1} P_n\left(\boldsymbol{r}^{A^T}\boldsymbol{r}_k\right) - \left(\frac{R}{r^B}\right)^{n+1} P_n\left(\boldsymbol{r}^{B^T}\boldsymbol{r}_k\right) \right\}$ |
| terrestrial and airborne | $B_r(\boldsymbol{x}, \boldsymbol{x}_k) = \sum_{n=0}^{\infty} \frac{2n+1}{4\pi} \left(-\frac{(n+1)}{r}\right) \left(\frac{R}{r}\right)^{n+1} B_n P_n(\boldsymbol{r}^T\boldsymbol{r}_k)$ |
| GOCE | $B_{rr}(\boldsymbol{x}, \boldsymbol{x}_k) = \sum_{n=0}^{\infty} \frac{2n+1}{4\pi} \frac{(n+1)(n+2)}{r^2} \left(\frac{R}{r}\right)^{n+1} B_n P_n(\boldsymbol{r}^T\boldsymbol{r}_k)$ |

**Table 2: Regularization parameter choice methods**

| Number | Method |
|---|---|
| 1) | 'the L-curve method based on CM 1' |
| 2) | 'VCE based on CM 1' |
| 3) | 'VCE based on CM 2' |
| 4) | 'VCE + L-curve method' |
| 5) | 'L-curve method + VCE' |

**Table 3: Study cases**

| Study Case | Data combination | Validation data |
|---|---|---|
| A | GRACE + Terrestrial I | 30'×30' highest degree 250 |
| B | GRACE + GOCE | |
| C | GOCE + Terrestrial I | |
| D | Terrestrial II + Airborne I | 5'×5' highest degree 2190 |
| E | GRACE + Terrestrial II + Airborne I | |
| F | GRACE + GOCE + Terrestrial II +Airborne I | |




**Table 4: Results of Study Case A: the RMS values (second column), the correlations between the estimated coefficients $d_k$ and the validation data (third column) as well as the correlations between the recovered gravity field and the validation data (forth column) for each regularization parameter choice method**

| | RMS $(m^2/s^2)$ | Correlation $d_k$ and $T_v$ | Correlation $T_c$ and $T_v$ |
|---|---|---|---|
| 'the L-curve method based on CM 1' | 4.4317 | 0.5578 | 0.9975 |
| 'VCE based on CM 1' | 4.4421 | 0.5590 | 0.9974 |
| 'VCE based on CM 2' | 3.7648 | 0.5598 | 0.9982 |
| 'VCE + L-curve method' | 3.6107 | 0.5687 | 0.9984 |
| 'L-curve method + VCE' | 3.6422 | 0.5636 | 0.9983 |

**Table 5: Results of Study Case F: the RMS values (second column), the correlations between the estimated coefficients $d_k$ and the validation data (third column) as well as the correlations between the recovered gravity field and the validation data (forth column) for each regularization parameter choice method**

| | RMS $(m^2/s^2)$ | Correlation $d_k$ and $T_v$ | Correlation $T_c$ and $T_v$ |
|---|---|---|---|
| 'the L-curve method based on CM 1' | 3.2876 | 0.8746 | 0.9950 |
| 'VCE based on CM 1' | 2.5510 | 0.6022 | 0.9957 |
| 'VCE based on CM 2' | 0.8282 | 0.9050 | 0.9996 |
| 'VCE + L-curve method' | 0.7837 | 0.9199 | 0.9996 |
| 'L-curve method + VCE' | 0.7983 | 0.9167 | 0.9996 |

**Table 6: RMS value results of each method for different study cases**

| Parameter choice method | A | B | C | D | E | F |
|---|---|---|---|---|---|---|
| 1) | 4.4317 | 7.4624 | 4.6946 | *1.0178* | 3.0465 | 3.2876 |
| 2) | 4.4421 | 14.230 | 4.7025 | 1.0325 | 3.1195 | 2.5510 |
| 3) | 3.7648 | 6.5635 | *3.2108* | 1.0446 | 0.8418 | 0.8282 |
| 4) | **3.6107** | **4.4325** | **3.2072** | **1.0148** | **0.8192** | **0.7837** |
| 5) | *3.6422* | *4.7446* | 3.2229 | 1.0317 | *0.8229* | *0.7983* |





**Table 7: Correlations between the estimated coefficients and the validation data of each method for different study cases**

| Parameter choice method | A | B | C | D | E | F |
|---|---|---|---|---|---|---|
| 1) | 0.5578 | 0.5435 | 0.5292 | *0.9172* | 0.8795 | 0.8746 |
| 2) | 0.5590 | 0.5070 | 0.5242 | *0.9172* | 0.7374 | 0.6022 |
| 3) | 0.5598 | 0.5336 | *0.5527* | 0.9169 | 0.8973 | 0.9050 |
| 4) | **0.5687** | **0.5617** | **0.5562** | **0.9173** | **0.9114** | **0.9199** |
| 5) | *0.5636* | *0.5572* | 0.5433 | 0.9169 | *0.9002* | *0.9167* |

**Table 8: RMS value results of each method for different study cases using the CuP function (as a comparison scenario w.r.t Table**
5 **6)**

| Parameter choice method | A | B | C | D | E | F |
|---|---|---|---|---|---|---|
| 1) | 5.1476 | 7.5402 | 6.5893 | *0.9545* | 3.1474 | 2.6667 |
| 2) | 4.5741 | 9.1576 | 6.1179 | 0.9768 | 2.6592 | 2.0826 |
| 3) | 3.9995 | *4.7468* | *3.4102* | 1.0088 | *0.7935* | 0.8102 |
| 4) | **3.6128** | **4.6946** | **3.0537** | **0.9541** | **0.7860** | **0.7795** |
| 5) | *3.9394* | 4.7735 | 3.4449 | 0.9963 | 0.8438 | *0.8084* |

**Table 9: Correlations between the estimated coefficients and the validation data of each method using the CuP function (as a comparison scenario w.r.t Table 7)**

| Parameter choice method | A | B | C | D | E | F |
|---|---|---|---|---|---|---|
| 1) | 0.4956 | 0.4752 | 0.3800 | *0.7954* | *0.6957* | *0.6955* |
| 2) | 0.4370 | 0.4709 | 0.3845 | 0.7922 | 0.3647 | 0.3950 |
| 3) | 0.5264 | 0.5165 | *0.5144* | 0.7892 | 0.5613 | 0.6031 |
| 4) | **0.5397** | **0.5344** | **0.5187** | **0.8045** | **0.7197** | **0.7679** |
| 5) | *0.5296* | *0.5237* | 0.5116 | 0.7824 | 0.6940 | 0.6190 |



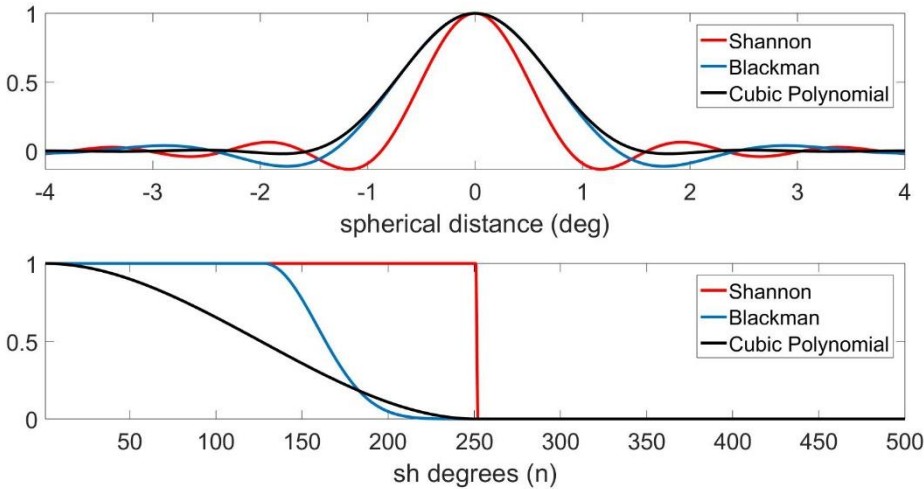

**Figure 1: Different SRBFs in the spatial domain (top, ordinate values are normed to 1) and the spectral domain (bottom) for $N_{max} = 255$.**

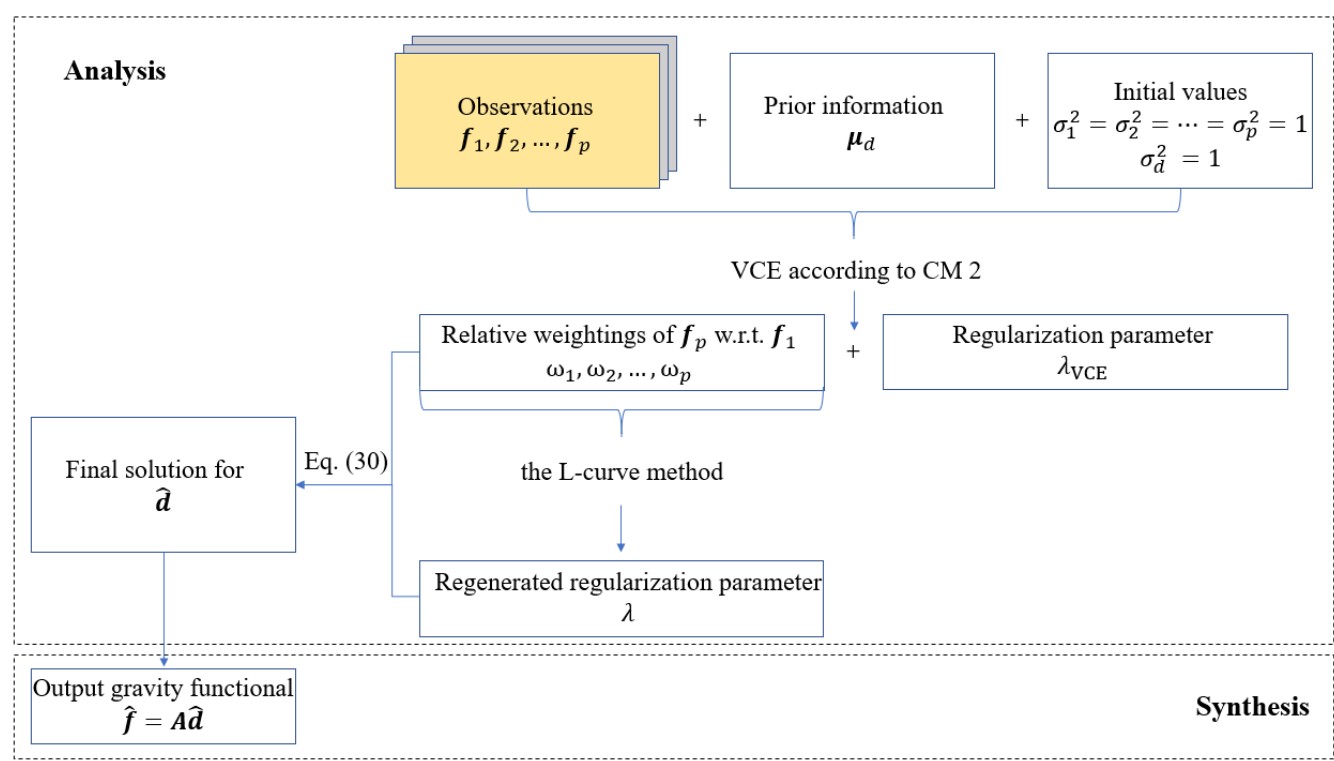

**Figure 2: Analysis and synthesis for combining different types of observations based on the 'VCE + L-curve method'**



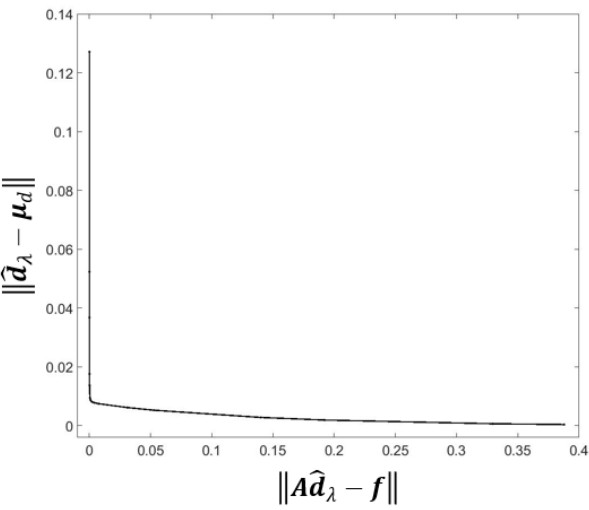

**Figure 3: The L-curve function**

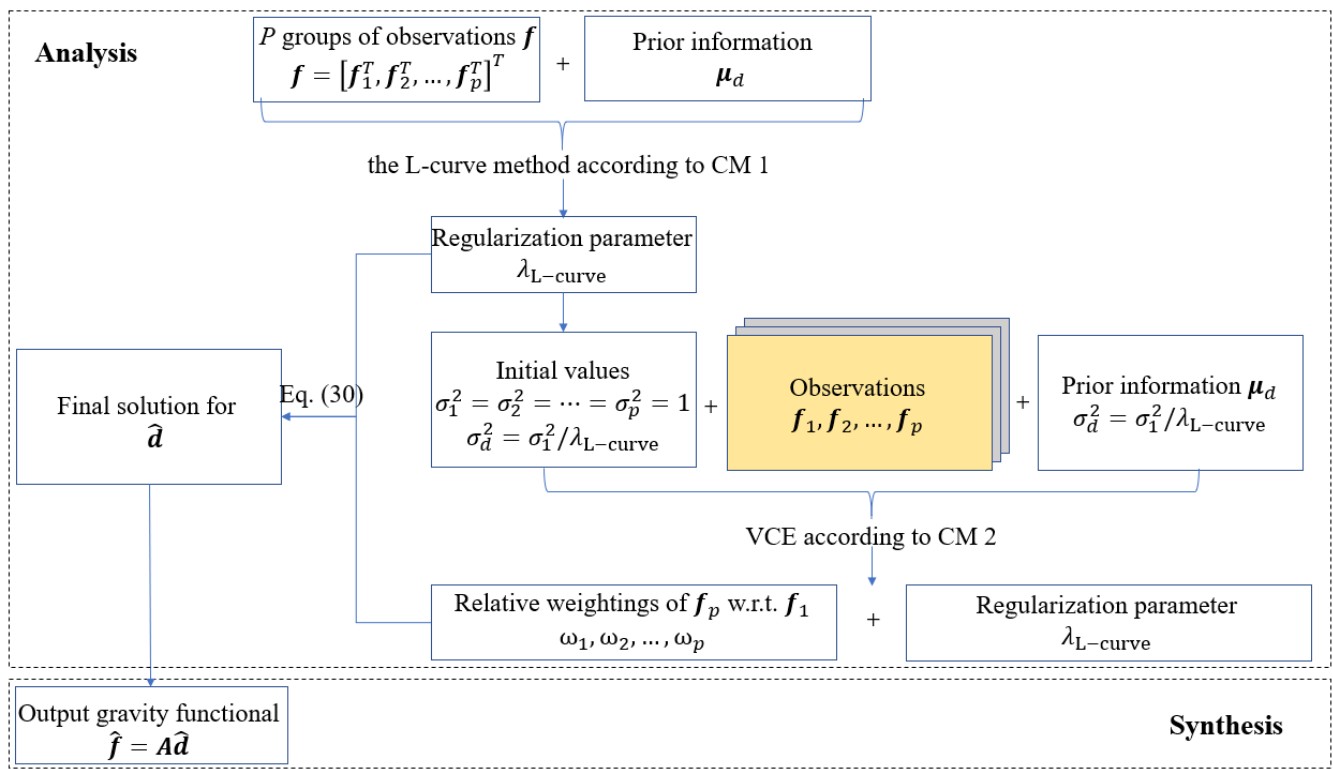

5     **Figure 4: Analysis and synthesis for combining different types of observations based on the 'L-curve method + VCE'**



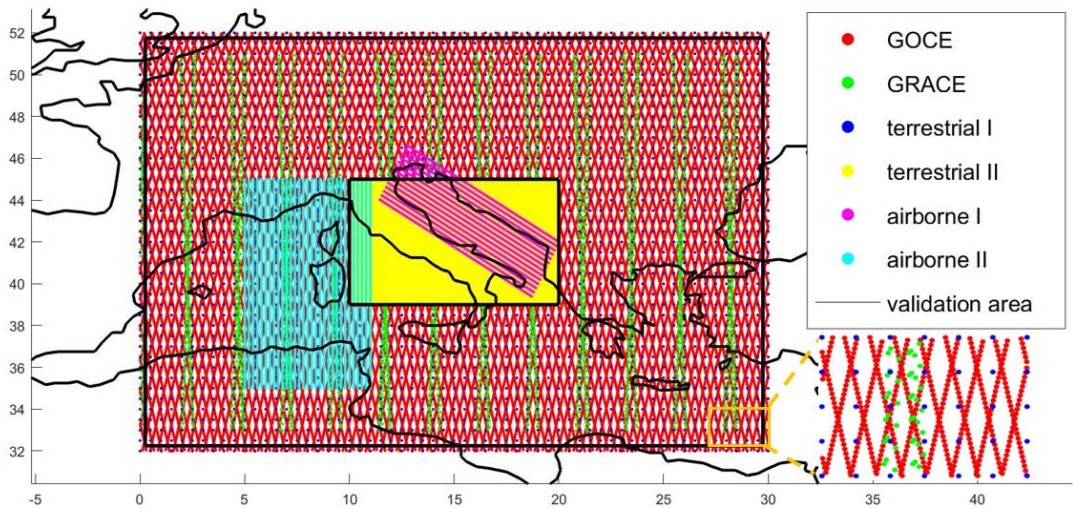

**Figure 5: Simulated GRACE, GOCE, terrestrial and airborne observations in 'Europe'**

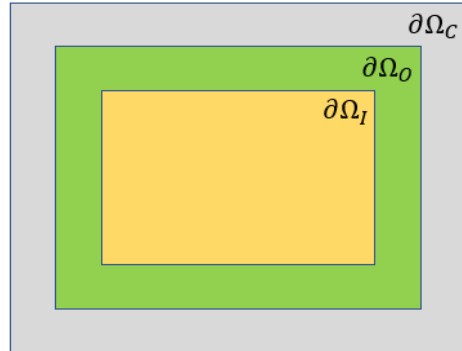

5    **Figure 6: Different extensions for the areas of computation $\partial\Omega_C$, of observations $\partial\Omega_O$ and of investigation $\partial\Omega_I$**



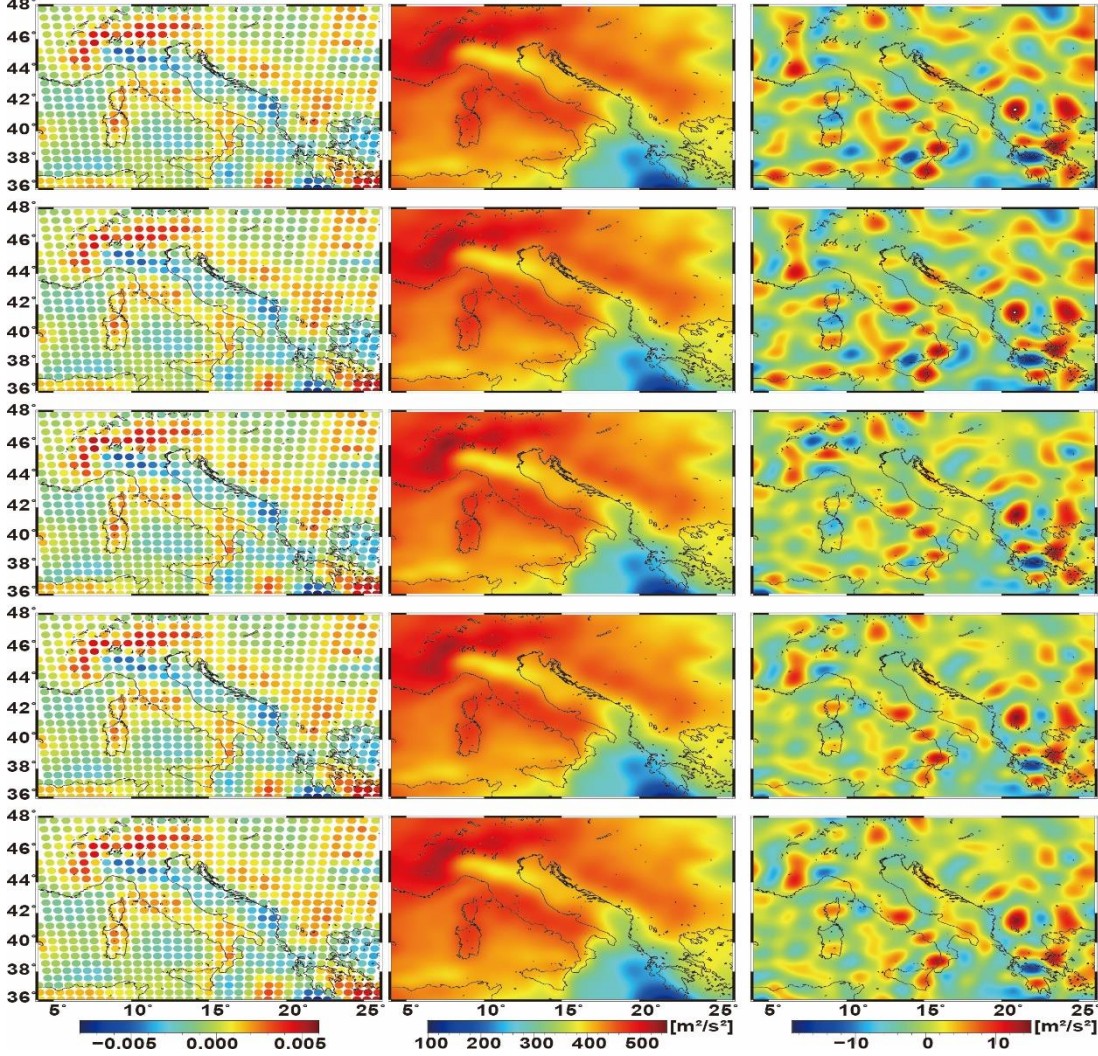

**Figure 7: The estimated coefficients $d_k$ (left column), the recovered disturbing potential (mid column) and the differences w.r.t the validation data (right column) for study case A. The results are obtained using regularization methods 'the L-curve method based on CM 1' (first row), 'VCE based on CM 1' (second row), 'VCE based on CM 2' (third row), 'VCE + L-curve method' (forth row) and 'L-curve method + VCE' (fifth row); see also Table 4 for numerical results.**





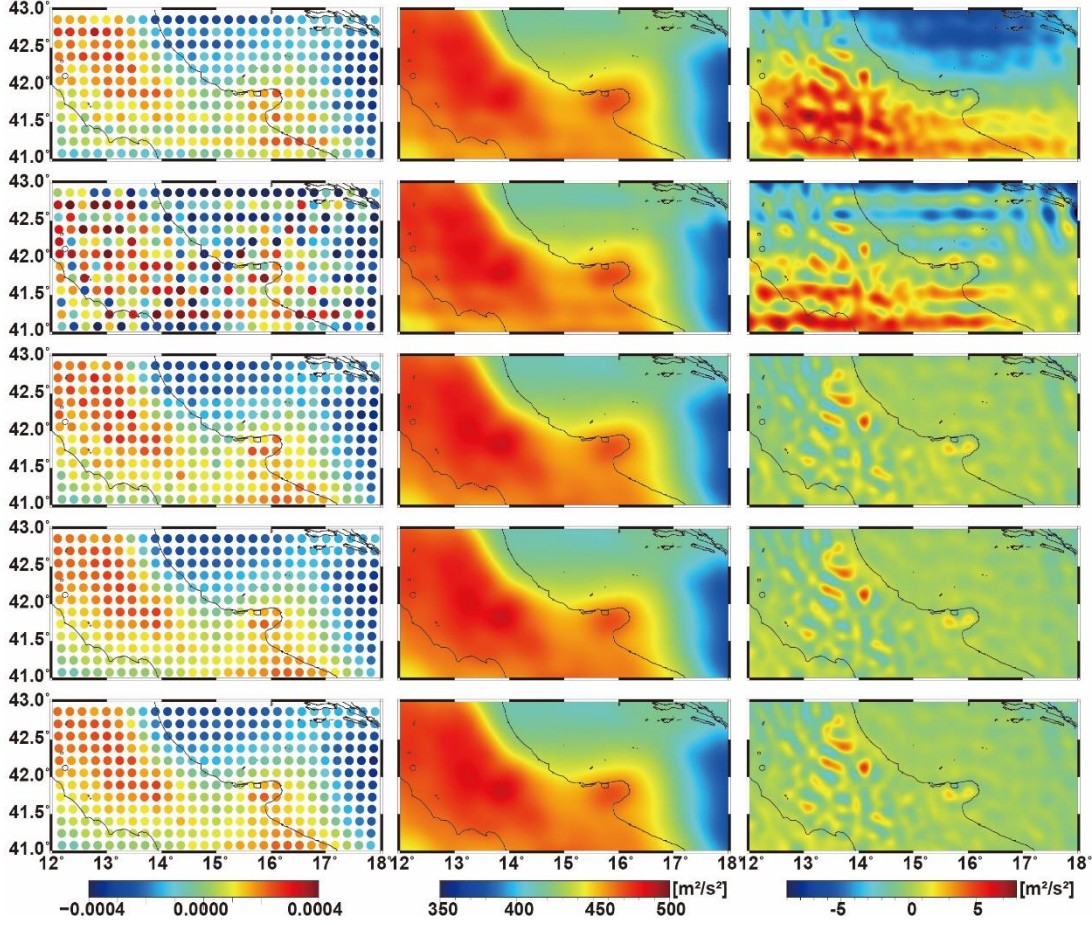

**Figure 8: The estimated coefficients $d_k$ (left column), the recovered disturbing potential (mid column) and the differences w.r.t the validation data (right column) for study case F. The results are obtained using regularization methods 'the L-curve method based on CM 1' (first row), 'VCE based on CM 1' (second row), 'VCE based on CM 2' (third row), 'VCE + L-curve method' (forth row) and 'L-curve method + VCE' (fifth row); see also Table 5 for numerical results.**