# Peer review of "Regularization methods for the combination of heterogeneous observations using spherical radial basis functions"

_Solid Earth, 2019_

## Referee Comment (RC1) · Anonymous Referee #1 · 30 Apr 2019

The manuscript is a good scientific writing, with clearly defined question, i.e. how to choose the optimal Tikhonov regularization parameter in the combination of heterogeneous observation for regional gravity field modeling. Numerical experiments are properly designed to investigate this problem and the results are fully discussed to get a reasonable conclusion. However, I couldn't see much scientific significance from the manuscript.

Among the five candidate regularization parameter choice methods, the first two methods that based on CM 1, i.e. equal weighting between different observation types, are naturally anticipated to be worse than the other three methods that based on CM 2 even

without numerical experiments. The proposed two new approaches, VCE+L-curve and L-curve+VCE, differs in fixing which relative weighting first, between different observation types or between observation and a-prior information, and seems there is not much scientific rationale in these two approaches. Moreover, the differences in residual RMS of the two approaches are not significant, just around 1-2%, the differences in correlations are even smaller.

Instead of the proposed two approaches, I prefer the third approach, i.e. VCE based on CM2, but with iteration. Since the relative weighting between different observations (weighting factors) and relative weighting between observations and a-prior information (regularization parameter) are adjusted simultaneously. Give the same initial variance components in the VCE+L-curve method, apply VCE based on CM2, then iterate this process with newly estimated variance components. I didn't find description on whether iteration is applied on the 'VCE based on CM2' approach in this manuscript, if not, I highly suggest the authors to test this iterated VCE method.

Also the 'regularization methods' in the title seems to be too 'big' for this manuscript, actually only one regularization method, Tikhonov regularization, is applied in this manuscript, something like 'regularization parameter choice methods' may be more appropriate.

Then there are some minor problems and typos:

Line 28, page 1, the fact that they are fulfilling the Laplacian ..., change to 'they fulfill...'.

Line 5, page 2, remove the acronym 'SRBF' since it's already defined in the abstract.

Line 6, page 3, ... the aforementioned two combination models, actually the CM1 and CM2 are not mentioned yet.

Line 3, page 5, ... the tensor $\Delta V$ of the gravity gradients $V_{ab}$, change to '... the tensor of the gravity potential ...'.

Equation 10, usually we use the symbol $\Delta$ to denote Laplacian operator in physical

geodesy, suggest to use $\nabla^2 V$ for tensor of gravity potential, and change $trace(\Delta V) = 0$ to $\Delta V = 0$.

Line 7 page 5, $V_{zz} \approx V_{rr}$, this doesn't stand universally, actually they are only approximately equal around the z-axis.

Typo in equation 25, the variances are all the same $\sigma_1^2 P_1^{-1}$, which should not.

Line 2, page 9, plotting the norm of ..., change to 'plotting the log of the norm of ...'.

Line 11, page 12, ... is chosen to 4 degree, change to '... is chosen to be 4 degree', same problem in line 1 page 13.

Figure 5, according to the legend, the blue dots represent terrestrial I data, but how can there be terrestrial data even over the Mediterranean sea? Moreover, I can't identify the validation area in the figure.

―――――――――――――――――――

---

## Referee Comment (RC2) · Anonymous Referee #2 · 15 May 2019

General comments: The authors present a regularization method that is used in the combination of heterogeneous data. The novelty is the combination of two regularization methods, namely by combining VCE estimation and the L-curve criteria. Various combinations are discussed and compared to existing methods, primarily to VCE estimation or L-curve regularization alone. However, the applied methodology is questionable as the combination of the two criteria is essentially equivalent to a double regularization. The comparison to the calculation "VCE based on CM2" reveals that the same result can be achieved by ordinary approaches. Further, the usage of the Shannon function for analysis and Blackman/CuP function results in additional smoothing which has not been further explained or described. Thus, I consider this paper inconclusive.

[Figure]

Specific comments: Section 1: The motivation of the regularization is unclear. Why are new methods needed? What are the limitations of existing methods? Why is the specific approach of the authors chosen and what benefits do the authors expect from their approach?

page 3, line 5: The authors argue to find the best-performing method (in what sense?) for regularization. However, they do not consider other methods than VCE and L-curve, e.g. GCV. Further, the method will be best-performing for their specific problem as no general criteria is derived which allows to conclude that the proposed method is best-performing.

Section 2.3: The authors present three different SRBFs with various smoothing features. Why is the approach of Eicker (2008) not considered? By including gravity field information into Bn, a considerable improvement can be achieved.

Section 2.3: If I understood the author's approach correctly, they use the Shannon function for the analysis of the simulated data but apply the estimated coefficients using either the Blackman or CuP function in the synthesis step. This approach is at least odd and inconsistent if not wrong from the beginning. In-fact, the approach introduces an additional smoothing. The authors state correctly that the latter two have smoothing features. Thus, the approach is unsuitable for the conducted research as it masks the effects of the regularization. It is another implicit regularization and thus the results cannot unambiguously assigned to the performance of the chosen methods. The only correct approach is therefore to use the same function for the analysis and synthesis step. The approach is even more questionable as Bentel2013 showed that differences between SRBFs matter (as also stated by the authors).

Section 3.1 provides no new information. The content can be reduced to the most significant equations and appropriate referencing.

Section 3.2: CM1 can obviously be removed as the assumption $\sigma_1^2 = \sigma_2^2 = ...$ is hardly valid in any case (except for simulated data with exactly
this assumption). Furthermore, applying VCE is the proper tool to consider data with varying variance factors. Thus, the results of CM1 are superfluous and the results prove the invalidity of the assumption.

Section 4.3: The regularization is essentially a double differentiation as the estimated variance factors during the VCE will reflect the regularization parameters. Practically the \lambda of equation 30 is split in \lambda_1 + \lambda_2 where one is estimated by VCE and the other by the L-curve criterion or vice-versa. Due to the double regularization, the results will be further smoothed than in case of applying just one of the methods alone. A better fit is therefore expected as the inherent effects due to ill-posedness is dominating. Also, the authors do not motivate the need for a second regularization and also do not discuss the effect of the second regularization step.

Section 5.3: The authors present two study cases: A and F; why not naming them A and B as you only present results of those two. The reader will have no information on cases B to E. Further, the results of CuP function can also be removed as they do not introduce any new insight.

---

## Author Comment (AC1) · 21 Jun 2019

Dear Reviewer,

Thank you very much for taking the time to review the manuscript and give comments that help us to improve it. Please find below our point-by-point response to your comments. The original comments are in black, and our answers are in blue. The revised manuscript with tracked changes is also attached.

Please also note the supplement to this comment:
https://www.solid-earth-discuss.net/se-2019-60/se-2019-60-AC1-supplement.pdf

[Figure]

[Figure]

**Supplement:**

**Dear Reviewer,**

Thank you very much for taking the time to review the manuscript and give comments that help us to improve it. Please find below our point-by-point response to your comments. The original comments are in black, and our answers are in blue. The revised manuscript with tracked changes is also attached.

The manuscript is a good scientific writing, with clearly defined question, i.e. how to choose the optimal Tikhonov regularization parameter in the combination of heterogeneous observation for regional gravity field modeling. Numerical experiments are properly designed to investigate this problem and the results are fully discussed to get a reasonable conclusion. However, I couldn't see much scientific significance from the manuscript.

As you said, this manuscript is trying to answer the question "how to choose the optimal Tikhonov regularization parameter in the combination of heterogeneous observation for regional gravity field modeling." Xu et al. (2006) pointed out that when heterogeneous measurements are involved, the conventional regularization parameter choice method, including the L-curve method and GCV may be not proper to apply. The reason is that the L-curve method or GCV cannot determine the relative weighting between different observation types. On the other hand, VCE can estimate the variance components of different observation techniques as well as the regularization parameter (interpreted as the ratio between the variance factors of the observations and the prior information) simultaneously. However, Naeimi (2013), Liang (2017) and Lieb (2017) all showed that VCE sometimes could not provide a reliable regularization result by incorporating prior information. Because in VCE, the prior information is interpreted as an additional observation technique, which is required to be stochastic. In regional gravity field modeling, usually a background model serves as the prior information, which is deterministic.

Based on these facts, the three commonly used regularization parameter choice methods seem all have pitfalls when heterogeneous observations are combined. Thus, we proposed two methods which combine VCE for determining the relative weights between the observation types and the L-curve method for determining the regularization parameter. The idea of combining VCE for weighting different data sets and a method for determining the regularization parameter was introduced in the Section 'future work' of both Naeimi (2013, p.121) and Liang (2017, p.134). The study

in this manuscript is also inspired by Wang et al. (2018), who combine two methods successively for determining the regularization parameter and relative weights for GPS and InSAR. However, to the best of our knowledge, there are still no publications applying this idea for combining heterogeneous observations in regional gravity field modeling.

Our experiment results prove that the original L-curve method or the VCE cannot always give a satisfying result (please refer to Table 6 in the revised manuscript), and our proposed 'VCE + L-curve method' provides the best results which means the smallest RMS error and the largest correlation for all our study cases. A further work based on the proposed 'VCE + L-curve method' for combining real observation data to model the regional gravity field in Colorado, USA, is under investigation. Moreover, this new approach could be used not only for regional gravity field modeling but also in many other fields such as atmospheric science.

We have rewritten parts of Section 1 and added two paragraphs explaining the limitations of the existing methods as well as the reasons for proposing the new methods. We have also highlighted the scientific significance of the new methods in the discussion part as well as the conclusion section.

Among the five candidate regularization parameter choice methods, the first two methods that based on CM 1, i.e. equal weighting between different observation types, are naturally anticipated to be worse than the other three methods that based on CM 2 even without numerical experiments. The proposed two new approaches, VCE+L-curve and L-curve+VCE, differs in fixing which relative weighting first, between different observation types or between observation and a-prior information, and seems there is not much scientific rationale in these two approaches. Moreover, the differences in residual RMS of the two approaches are not significant, just around 1-2%, the differences in correlations are even smaller.

We have removed the method 'VCE based on CM 1' since it is expected to perform worse than 'VCE based on CM 2'. However, due to the aforementioned reason, VCE cannot generate a reliable regularization parameter. In this case, VCE based on CM 2 sometimes gives even worse results than the L-curve method based on CM 1. Please refer to the results of study cases B and D in Table 6 and Table 7 of the revised manuscript.

We have rewritten the result section, and the comparison between 'VCE based on CM 1' and 'VCE based on CM 2' is removed. Now we focus on the comparison of the two proposed methods to the original L-curve method and VCE.

We have added the scientific rationale for the proposed two new approaches, which is to use VCE for determining the relative weights between different observation types and the L-curve method for determining the regularization parameter. The two methods differ in the order of the applied procedures, i.e., applying VCE or the L-curve method first. The improvements in the RMS values of 'VCE + L-curve method' compare to 'Lcurve method + VCE' with respect to the validation data are generally not significant when the Shannon function is used, except for case study B where the difference reaches 6.6%. However, when the CuP function is used, the RMS errors of 'VCE + Lcurve' decrease 8.3%, 1.7%, 11.4%, 4.2%, 6.8%, 3.6% (see Table 8) compared to 'Lcurve + VCE' in the six study cases, respectively. Thus, we conclude that both newly proposed approaches are performing well, but 'VCE + L-curve' is more stable regarding different type of SRBFs (see Page 14, Line 5-7). More important, 'VCE + L-curve method' outperforms the original VCE in all the study cases no matter using a smoothing or non-smoothing SRBF; 'L-curve method + VCE' outperforms VCE when the Shannon function is used, but does not show significant improvements compared to the VCE when the CuP is used.

Instead of the proposed two approaches, I prefer the third approach, i.e. VCE based on CM2, but with iteration. Since the relative weighting between different observations (weighting factors) and relative weighting between observations and a-prior information (regularization parameter) are adjusted simultaneously. Give the same initial variance components in the VCE+L-curve method, apply VCE based on CM2, then iterate this process with newly estimated variance components. I didn't find description on whether iteration is applied on the 'VCE based on CM2' approach in this manuscript, if not, I highly suggest the authors to test this iterated VCE method.

We would like to clarify that the 'VCE based on CM 2' approach presented in the manuscript has already included the iteration procedure, and the same initial values were given to both the 'VCE based on CM 2' and 'VCE + L-curve method'. We have mentioned in Page 9, Line 9 that we estimate the variance components within an iterative process.

We have added more explanations to Section 4.2 showing why VCE could lead to unreliable regularization results.

Also the 'regularization methods' in the title seems to be too 'big' for this manuscript, actually only one regularization method, Tikhonov regularization, is applied in this manuscript, something like 'regularization parameter choice methods' may be more appropriate.

We have changed the title to 'Regularization parameter choice methods for the combination of heterogeneous observations using spherical radial basis functions'.

Then there are some minor problems and typos:

Line 28, page 1, the fact that they are fulfilling the Laplacian ..., change to 'they fulfill...'.

We have changed this sentence as suggested.

Line 5, page 2, remove the acronym 'SRBF' since it's already defined in the abstract.

We have removed it.

Line 6, page 3, ... the aforementioned two combination models, actually the CM1 and CM2 are not mentioned yet.

The two combination models have been mentioned and introduced on Page 2, Line 18-20. However, this whole paragraph has been changed in the revised version corresponding to the first comment.

Line 3, page 5, ... the tensor  $\Delta V$  of the gravity gradients  $V_{ab}$ , change to '... the tensor of the gravity potential ...'.

We have changed the sentence to '...observed the gravity gradients  $V_{ab}$  with  $a, b \in \{x, y, z\}$ , i.e. all second-order derivatives of the gravitational potential V...'

Equation 10, usually we use the symbol  $\Delta$  to denote Laplacian operator in physical geodesy, suggest to use  $\nabla^2 V$  for tensor of gravity potential, and change trace  $(\Delta V) = 0$  to  $\Delta V = 0$ .

To keep the symbol consistent, we have changed Eq. (10) to  $V_{ab} = \begin{bmatrix} V_{xx} & V_{xy} & V_{xz} \\ V_{yx} & V_{yy} & V_{yz} \\ V_{zx} & V_{zy} & V_{zz} \end{bmatrix}$ , and changed trace (AV) = 0 to trace  $(V_{xy}) = 0$ .

and changed trace  $(\Delta V) = 0$  to trace  $(V_{ab}) = 0$ .

Line 7 page 5,  $V_{zz} \approx V_{rr}$ , this doesn't stand universally, actually they are only approximately equal around the z-axis.

We have changed this sentence to 'The observation data of GOCE used in this study are simulated as the radial component  $V_{rr} = \frac{\partial^2 V}{\partial r^2}$ , and the observation equation reads...'.

Typo in equation 25, the variances are all the same  $\sigma_1^2 P_1^{-1}$ , which should not.

| This is a typo, we have changed it to D | $\begin{pmatrix} \begin{bmatrix} \boldsymbol{f}_1 \\ \boldsymbol{f}_2 \\ \vdots \\ \boldsymbol{f}_p \\ \boldsymbol{\mu}_d \end{bmatrix} =$ | $= \begin{bmatrix} \sigma_1^2 \boldsymbol{P}_1^{-1} \\ \boldsymbol{0} \\ \vdots \\ \vdots \\ \boldsymbol{0} \end{bmatrix}$ | $     \begin{array}{c}       0 \\       \sigma_2^2 P_2^{-1} \\       0 \\       \vdots \\       0     \end{array} $ | 0
::
::
:: | $ \begin{matrix} \vdots \\ \vdots \\ \sigma_p^2 \boldsymbol{P}_p^{-1} \\ \boldsymbol{0} \end{matrix} $ | $\begin{bmatrix} 0 \\ \vdots \\ \vdots \\ 0 \\ \sigma_d^2 \boldsymbol{P}_d^{-1} \end{bmatrix}$ |
|-----------------------------------------|--------------------------------------------------------------------------------------------------------------------------------------------|----------------------------------------------------------------------------------------------------------------------------|---------------------------------------------------------------------------------------------------------------------|---------------------|--------------------------------------------------------------------------------------------------------|------------------------------------------------------------------------------------------------|
|-----------------------------------------|--------------------------------------------------------------------------------------------------------------------------------------------|----------------------------------------------------------------------------------------------------------------------------|---------------------------------------------------------------------------------------------------------------------|---------------------|--------------------------------------------------------------------------------------------------------|------------------------------------------------------------------------------------------------|

Line 2, page 9, plotting the norm of ..., change to 'plotting the log of the norm of ...'. Done. Line 11, page 12, ... is chosen to 4 degree, change to '... is chosen to be 4 degree', same problem in line 1 page 13.

**We have changed both of them.**

Figure 5, according to the legend, the blue dots represent terrestrial I data, but how can there be terrestrial data even over the Mediterranean sea? Moreover, I can't identify the validation area in the figure.

Parts of the terrestrial I data are over the Mediterranean sea because these data are not real observations but simulated from the EGM2008.

For clarification we have added a sentence to Line 23, page 10, "The two validation areas are presented in Figure 5 with black rectangles."

**References:**

Koch, K. R., and Kusche, J.: Regularization of geopotential determination from satellite data by variance components, Journal of Geodesy, 76(5), 259-268, doi: 10.1007/s00190-002-0245-x, 2002.

Liang, W.: A regional physics-motivated electron density model of the ionosphere, PhD thesis, German Geodetic Research Institute, Technical University of Munich, Germany, 2017.

Lieb V.: Enhanced regional gravity field modeling from the combination of real data via MRR, PhD thesis, German Geodetic Research Institute, Technical University of Munich, Germany, 2017.

Naeimi, M.: Inversion of satellite gravity data using spherical radial base functions, PhD thesis, Institute of Geodesy, University of Hanover, Germany, 2013.

Wang, L., Zhao, X., and Gao, H.: A method for determining the regularization parameter and the relative weight ratio of the seismic slip distribution with multi-source data, Journal of Geodynamics, 118, 1-10, doi:10.1016/j.jog.2018.04.005, 2018.

Xu, P., Shen, Y., Fukuda, Y., and Liu, Y.: Variance component estimation in linear inverse ill-posed models, Journal of Geodesy, 80(2), 69-81, doi:10.1007/s00190-006-0032-1, 2006.

From 21 Jun 2019 to 04 Dec 2019 a revised version of the manuscript including track changes was available in this supplement. Upon the author's request it was removed.

---

## Author Comment (AC2) · 21 Jun 2019

Dear Reviewer,

Thank you very much for taking the time to review the manuscript and give comments that help us to improve it. Please find below our point-by-point response to your comments. The original comments are in black, and our answers are in blue. The revised manuscript with tracked changes is also attached.

Please also note the supplement to this comment:
https://www.solid-earth-discuss.net/se-2019-60/se-2019-60-AC2-supplement.pdf

[Figure]

**Supplement:**

**Dear Reviewer,**

Thank you very much for taking the time to review the manuscript and give comments that help us to improve it. Please find below our point-by-point response to your comments. The original comments are in black, and our answers are in blue. The revised manuscript with tracked changes is also attached.

General comments: The authors present a regularization method that is used in the combination of heterogeneous data. The novelty is the combination of two regularization methods, namely by combining VCE estimation and the L-curve criteria. Various combinations are discussed and compared to existing methods, primarily to VCE estimation or L-curve regularization alone. However, the applied methodology is questionable as the combination of the two criteria is essentially equivalent to a double regularization. The comparison to the calculation "VCE based on CM2" reveals that the same result can be achieved by ordinary approaches. Further, the usage of the Shannon function for analysis and Blackman/CuP function results in additional smoothing which has not been further explained or described. Thus, I consider this paper inconclusive.

We do not apply a double regularization. We combine the VCE for determining the relative weighting between different observation types and the L-curve method for determining the regularization parameter. From the VCE procedure, only the relative weights are kept, the generated regularization parameter is not further used.

The experiments show that 'VCE based on CM 2' cannot guarantee a reliable result. When the Shannon function is used for both analysis and synthesis as suggested, 'VCE based on CM 2' gives even worse results (larger RMS errors as well as smaller correlations) than 'the L-curve method based on CM 1' in the study cases B and D (please refer to Table 6 in the revised manuscript). The 'VCE + L-curve method' overperforms 'VCE based on CM 2' by 6.1%, 48%, 0.7%, 4.1%, 2.3% and 1.8% in terms of RMS error in the six study cases, respectively. When the CuP function is used, 'VCE based on CM 2' gives smaller correlation results than the 'L-curve method based on CM 1' in three study cases. 'VCE + L-curve method' overperforms 'VCE based on CM 2' gives smaller correlation results than the 'L-curve method based on CM 1' in three study cases. 'VCE + L-curve method' overperforms 'VCE based on CM 2' by 9.7\%, 1.1%, 10.5\%, 5.4\%, 0.9\%, 3.8\%, respectively. More important, the performance of 'VCE + L-curve method' is stable in all the study cases. Thus, we think the improvement by using our proposed method is in fact significant.

The idea to use certain spherical radial basis function for the analysis and the same or

other functions for the synthesis in case of a band limitation is explained by Schreiner (1996), Schmidt et al. (2007, Theorem 1) and Lieb (2017). However, since our goal is to compare different types of regularization parameter choice methods but not different SRBFs, we have changed the experiments to using the same SRBF for both analysis and synthesis. Thus, we have modified Section 2.3, and we have updated the results and discussions in Section 5.3.

Specific comments: Section 1: The motivation of the regularization is unclear. Why are new methods needed? What are the limitations of existing methods? Why is the specific approach of the authors chosen and what benefits do the authors expect from their approach?

We have replaced the content of Page 2, Line 29 to Page 3, Line 9 in Section 1 by the following two paragraphs to explain the motivation of this study in more detail. In the revised manuscript, we explain the limitations of using the L-curve method or the VCE alone, and why we proposed the two new methods.

"VCE estimates the variance components of different observation techniques as well as the regularization parameter simultaneously. However, in this case, the regularization parameter is handled as another variance component, and the prior information is interpreted as an additional observation technique, and, thus, assumed to be of random character. In most of the regional gravity modeling studies, a background model serves as prior information. In this case, the prior information has no random character, and the regularization parameter generated by VCE is not reliable (Liang, 2017). Lieb (2017, p.131) presents a case which shows the instability of VCE. Naeimi (2013, p.102) shows that VCE generally performs worse than the L-curve method.

Since VCE does not guarantee a reliable regularization solution, and the L-curve method cannot weight heterogeneous observations, the purpose of this paper is to combine these two methods, and to improve the stability and reliability of the solutions. The idea of combining VCE for weighting different data sets and a method for determining the regularization parameter was introduced in the Section 'future work' of both Naeimi (2013, p.121) and Liang (2017, p.134). The study in this manuscript is also inspired by Wang et al. (2018), who combine two methods successively for determining the regularization parameter and relative weights for GPS and InSAR. However, to the best of our knowledge, there are still no publications applying this idea for combining heterogeneous observations in regional gravity field modeling. Thus, we introduce and discuss in the paper the two proposed new methods which combine VCE for determining the relative weighting between different observation types and the Lcurve method for determining the regularization parameter, denoted as 'VCE + L-curve method' and 'L-curve method + VCE', depending on the order of the applied procedures. Numerical experiments are carried out to compare their performance to the original L-curve method and VCE."

page 3, line 5: The authors argue to find the best-performing method (in what sense?) for regularization. However, they do not consider other methods than VCE and L-curve,

e.g. GCV. Further, the method will be best-performing for their specific problem as no general criteria is derived which allows to conclude that the proposed method is best-performing.

We agree that the word "best-performing" is too "big" for this paper. The purpose of this paper is to test if the two combined methods give better results compared to VCE or the L-curve method alone. Since the L-curve method or other conventional regularization parameter choice methods cannot weight heterogeneous observations, and VCE does not guarantee a reliable regularization solution, we want to improve the stability and reliability of the solutions by combining these two methods. The criteria used for comparing the performance are the RMS error as well as the correlation between the estimated coefficients and the validation data (see Section 5.2, Page 11, Line 20-29).

We have already changed the whole paragraph corresponding to the first specific comment and have rewritten the purpose of the paper. Please refer to the answer of the last comment and the revised manuscript.

Section 2.3: The authors present three different SRBFs with various smoothing features. Why is the approach of Eicker (2008) not considered? By including gravity field information into Bn, a considerable improvement can be achieved.

The focus of this work is not to compare the performance of different SRBFs, but to compare the performance of different regularization parameter choice methods. For each group of comparison, the same SRBF is used for every regularization parameter choice method. However, as mentioned in the future work, we plan to study the performance of more types of SRBFs using the newly devised method.

Section 2.3: If I understood the author's approach correctly, they use the Shannon function for the analysis of the simulated data but apply the estimated coefficients using either the Blackman or CuP function in the synthesis step. This approach is at least odd and inconsistent if not wrong from the beginning. In-fact, the approach introduces an additional smoothing. The authors state correctly that the latter two have smoothing features. Thus, the approach is unsuitable for the conducted research as it masks the effects of the regularization. It is another implicit regularization and thus the results cannot unambiguously assigned to the performance of the chosen methods. The only correct approach is therefore to use the same function for the analysis and synthesis step. The approach is even more questionable as Bentel2013 showed that differences between SRBFs matter (as also stated by the authors).

In the previous version of the manuscript, we have conducted two sets of experiments; the first set uses the Shannon function for analysis and the Blackman function for synthesis; the second set uses both the CuP function for analysis and synthesis. Schreiner (1996) and Freeden et al. (1998) gave the proof that different types of SRBFs can be used in the analysis step and synthesis step in case of the same band limitation.

This procedure was applied in Lieb et al. (2016), Lieb (2017), among many others. However, since our goal is to compare different types of regularization parameter choice methods but not different SRBFs, we have changed the first set of experiments to use the Shannon function for both analysis and synthesis. Thus, we have removed the Blackman function and modified Section 2.3, consequently, we have also updated the RMS results for the first set of experiments and the corresponding discussions in Section 5.3.

Section 3.1 provides no new information. The content can be reduced to the most significant equations and appropriate referencing.

We have shortened some of the content. Since Section 3.1 is not very long, and it discusses how the coefficients are estimated and how the regularization parameter is introduced, we have kept significant equations.

Section 3.2: CM1 can obviously be removed as the assumption  $sigma_1^2 = sigma_2^2 = ...$  is hardly valid in any case (except for simulated data with exactly this assumption). Furthermore, applying VCE is the proper tool to consider data with varying variance factors. Thus, the results of CM1 are superfluous and the results prove the invalidity of the assumption.

The ordinary L-curve method can only be applied based on CM 1 because it cannot estimate the variance factors. And the results show that although results based on CM 1 are expected to be worse than those based on CM 2, 'the L-curve method based on CM 1' still performs better than 'VCE based on CM 2' in some study cases. So, the results of CM 1 prove that VCE does not guarantee reliable regularization results, and thus show the importance of our combined method.

However, we have removed the method 'VCE based on CM 1' since the variance factors need to be considered for different data sets, and the results from 'VCE based on CM 1' are expected and proved to be worse than 'VCE based on CM 2'. Section 5.3 and 6 are thus rewritten.

Section 4.3: The regularization is essentially a double differentiation as the estimated variance factors during the VCE will reflect the regularization parameters. Practically the \lambda of equation 30 is split in \lambda\_1 + \lambda\_2 where one is estimated by VCE and the other by the L-curve criterion or vice-versa. Due to the double regularization, the results will be further smoothed than in case of applying just one of the methods alone. A better fit is therefore expected as the inherent effects due to ill-posedness is dominating. Also, the authors do not motivate the need for a second regularization and also do not discuss the effect of the second regularization step.

We do not apply a double regularization. We used VCE for determining the relative weight between each observation types and the L-curve method for determining the regularization parameter. The regularization parameter that was generated from VCE is

not further used. The  $\lambda$  of Eq. (30) is only estimated by the L-curve criterion, and the relative weights  $\omega_p$  in Eq. (30) are estimated by VCE. For clarification, we have extended the description part in Section 4.3 in the revised paper.

Section 5.3: The authors present two study cases: A and F; why not naming them A and B as you only present results of those two. The reader will have no information on cases B to E. Further, the results of CuP function can also be removed as they do not introduce any new insight.

The naming depends on how many types of observations are combined; from A to F, more types of observations are combined. However, we have rewritten Section 5.3, because we have changed the first set of experiments to using the Shannon function for both analysis and synthesis, and the results from 'VCE based on CM 1' are removed. In the revised version, we discuss all the cases together, and Section 5.3 is rewritten. We think the results of the CuP function are necessary for two reasons. The first reason is that Naeimi (2013, p. 121) points out that VCE gives better performance when smoothing kernels which have built-in regularization are used. Our results of the CuP function show that even with a built-in regularization, VCE still does not guarantee a reliable result, and 'VCE + L-curve method' outperforms VCE in all the study cases. The second reason is that when the Shannon function is used for both analysis and synthesis, 'VCE + L-curve method' always outperforms the original L-curve method and VCE, and 'L-curve method + VCE' also generally outperforms the L-curve method and VCE. But when the CuP function is used, 'VCE + L-curve method' still performs the best but 'L-curve method + VCE' does not show significant improvements compared to VCE. So, we conclude that the 'VCE + L-curve method' improves the stability and reliability of the solution no matter the used SRBFs have a smoothing feature or not.

References:

Freeden W., Gervens T., Schreiner M.: Constructive Approximation on the Sphere (With Applications to Geomathematics), Oxford Science Publications, Clarendon Press, ISSN 0198536828, ISBN 978-0-19-853682-6, 1998.

Liang, W.: A regional physics-motivated electron density model of the ionosphere, PhD thesis, German Geodetic Research Institute, Technical University of Munich, Germany, 2017.

Lieb V.: Enhanced regional gravity field modeling from the combination of real data via MRR, PhD thesis, German Geodetic Research Institute, Technical University of Munich, Germany, 2017.

Naeimi, M.: Inversion of satellite gravity data using spherical radial base functions, PhD thesis, Institute of Geodesy, University of Hanover, Germany, 2013.

Schmidt, M., Fengler, M., Mayer-Gürr, T., Eicker, A., Kusche, J., Sánchez, L., and Han, S. C.: Regional gravity modeling in terms of spherical base functions, Journal of

Geodesy, 81(1), 17-38, doi:10.1007/s00190-006-0101-5, 2007.

Schreiner M.: A Pyramid Scheme for Spherical Wavelets, AGTM-Report, 170, Universität Kaiserslautern, doi:41A58. 42C15.44A35.65D15, 1996.

Wang, L., Zhao, X., and Gao, H.: A method for determining the regularization parameter and the relative weight ratio of the seismic slip distribution with multi-source data, Journal of Geodynamics, 118, 1-10, doi:10.1016/j.jog.2018.04.005, 2018.

From 21 Jun 2019 to 28 Nov 2019 a revised version of the manuscript including track changes was available in this supplement. Upon the authors' request it was removed.

---

## Editor Comment (EC1) · Nicolas Gillet (Editor) · 24 Jun 2019

Dear referee, the authors have now given their point by point answer to your comments, and provide a modified manuscript. could you let me know if, to your opinion, they have convincingly addressed your points or not ? best regards, Nicolas Gillet.